# A BDNF-TrkB autocrine loop enhances senescent cell viability

Carlos Anerillas [1] ✉, Allison B. Herman[1], Rachel Munk[1], Amanda Garrido [2], Kwan-Wood Gabriel Lam [1], Matthew J. Payea[1], Martina Rossi [1], Dimitrios Tsitsipatis[1], Jennifer L. Martindale [1], Yulan Piao[1], Krystyna Mazan-Mamczarz[1], Jinshui Fan[1], Chang-Yi Cui[1], Supriyo De[1], Kotb Abdelmohsen [1], Rafael de Cabo [2] & Myriam Gorospe [1] ✉

Cellular senescence is characterized by cell cycle arrest, resistance to apoptosis, and a senescence-associated secretory phenotype (SASP) whereby cells secrete pro-inflammatory and tissue-remodeling factors. Given that the SASP exacerbates age-associated pathologies, some aging interventions aim at selectively eliminating senescent cells. In this study, a drug library screen uncovered TrkB (NTRK2) inhibitors capable of triggering apoptosis of several senescent, but not proliferating, human cells. Senescent cells expressed high levels of TrkB, which supported senescent cell viability, and secreted the TrkB ligand BDNF. The reduced viability of senescent cells after ablating BDNF signaling suggested an autocrine function for TrkB and BDNF, which activated ERK5 and elevated BCL2L2 levels, favoring senescent cell survival. Treatment with TrkB inhibitors reduced the accumulation of senescent cells in aged mouse organs. We propose that the activation of TrkB by SASP factor BDNF promotes cell survival and could be exploited therapeutically to reduce the senescent-cell burden.

The cell senescence program is implemented in response to different forms of non-lethal cell damage. Senescent cells are characterized by several traits, including morphological changes, indefinite growth arrest, and metabolic reprogramming that includes altered lysosomal function with increased presence of senescence-associated β-Galactosidase (SA-β-Gal) activity[1]. However, one of their most prominent features is the senescence-associated secretory phenotype (SASP), a trait defined as the active secretion of pro-inflammatory factors and matrix-remodeling enzymes. Through these secreted proteins, senescent cells modify their environment and influence physiologic and disease processes[2].

On the one hand, senescence has been identified as critically beneficial for tissue repair, embryonic development, and tumor suppression; on the other, the uncontrolled accumulation of senescent cells within organs can lead to tissue dysfunction and disease[3]. With advancing age, senescent cells increase in tissues and organs, exacerbating a range of aging-related physiologic declines and diseases[4]. The pharmacologic removal of senescent cells using senolytics in mouse models was recently found to improve several age-associated disorders[5]. Many senolytic strategies exploit the fact that senescent cells exist in a state of potent suppression of apoptosis[6], for instance, by inhibiting the pro-survival effect of anti-apoptotic proteins in the BCL2 family[7,8].

A range of roles have been proposed for mitogen-activated protein kinase (MAPK) signaling pathways in senolysis[9]. To address the influence of MAPKs experimentally and systematically, we employed a chemical library including inhibitors of every step in the MAPK cascade, from the receptors down to the final effector proteins. This survey identified two drugs, both inhibitors of tropomyosin receptor kinase (Trk) activity, that effectively and specifically reduced the

[1]Laboratory of Genetics and Genomics, National Institute on Aging Intramural Research Program, National Institutes of Health, Baltimore, MD, USA. [2]Translational Gerontology Branch, National Institute on Aging Intramural Research Program, National Institutes of Health, Baltimore, MD, USA. ✉e-mail: carlos.anerillasaljama@nih.gov; myriam-gorospe@nih.gov

viability of senescent WI-38 and BJ human diploid fibroblasts. Trk receptors are best known for transducing signals from their known ligands, the neurotrophins, to modulate neuronal development and communication[10]. Among the Trk family, which includes TrkA, TrkB, and TrkC, we identified TrkB as being primarily responsible for maintaining senescent cell viability in this unexpected context. Further interrogation revealed that cells rendered senescent in different ways secrete the neurotrophin BDNF (brain-derived neurotrophic factor), and that in a range of senescence models, autocrine/paracrine activation of TrkB by BDNF sustained ERK5 activation and BCL2L2-dependent viability of senescent cells. The discovery that the early commitment to senescence required a rise in *BDNF* mRNA levels suggested that BDNF production helps to adjust the cellular outcome to the extent of damage received. Finally, in light of evidence that inhibiting this regulatory paradigm reduces the senescent cell burden in mouse tissues, we propose that suppressing the signaling through TrkB and BDNF can be exploited to eliminate senescent cells for therapeutic benefit.

## Results

### MAPK-directed drug screen identifies inhibitors of Trk receptors as potential senolytic drugs

As discussed recently[9], the role of MAPK pathways in senescence-associated apoptosis is poorly understood. To better understand how targeting MAPKs in senescence might offer therapeutic advantage, we screened a custom library (Tocris) comprising 43 compounds that inhibit different mediators of MAPK signaling cascades (Fig. 1a, **Source Data**). The screen included a known senolytic, ABT-737, as a positive control, and was carried out in parallel on proliferating cells to ensure that the identified drugs selectively affected senescent cells. To strengthen our screen, we performed it on two human diploid fibroblast lines (WI-38 and BJ) that are well-established models for senescence studies. The experimental conditions were previously optimized to trigger cellular senescence in both cell types with minimal cell death[11]. Briefly, WI-38 and BJ fibroblasts were treated with 50 or 25 μM etoposide, respectively, refreshing the drug every three days. After ten days, when cells had reached senescence, etoposide was removed, and the different compounds from the library were tested for an additional 48 h (Fig. 1b). The induction of senescence in WI-38 and BJ cells was confirmed by measuring senescence-associated β-Galactosidase (SA-β-Gal) activity (Supplementary Fig. 1a–d); growth arrest in the senescent cultures was confirmed by measuring BrdU incorporation (Supplementary Fig. 1e, f).

Next, we assessed the effect of treatment with the drugs in the library (each at 10 μM) for 48 h on both proliferating and senescent WI-38 or BJ cells. The impact of each drug was evaluated by direct cell counts at the end of the treatment relative to the initial cell numbers. As shown in Fig. 1c (further information in **Source Data**), treatment with several inhibitors caused specific decreases in the number of live senescent cells; the heat map (Fig. 1d) represents the percentage of senolysis caused by a certain drug (proportion of dead cells 48 h later). The drugs were grouped by the target proteins within the MAPK pathways upon which they act. We excluded from the heat map those drugs that decreased the viability of proliferating cells, such as CMPD-1 (an inhibitor of MK2 and microtubular assembly), BRD 7389 (a p90 S6K inhibitor), HI TOPK 032 (a TOPK inhibitor), and AC 710 (a PDGFR inhibitor). Strikingly, two of the drugs with the most pronounced senolytic effect were the TrkB receptor inhibitors GNF 5837 and ANA 12 (blue arrows in Fig. 1c, d). TrkB belongs to a family of Receptor Tyrosine Kinases (RTKs) whose ligands are neurotrophins, a group of secreted proteins crucial for engaging neuronal survival[12]. Although GNF 5837 also inhibits TrkA and TrkC, other inhibitors included in the library selectively targeting other members of this family, such as TrkA (GW 441756) and NGFR (PD 90780), did not show any senolytic properties. Thus, in this screen, TrkB appeared to be the main Trk

receptor responsible for maintaining the viability of senescent cells. Together, these data indicate that TrkB inhibitors have potential senolytic effects.

### Pharmacological inhibition of TrkB selectively induces apoptotic cell death of senescent cells

We sought to explore further the ability of Trk inhibitors to selectively induce apoptosis in senescent cells. In addition to GNF 5837 and ANA 12, our analysis included PF 06273340, an additional Trk inhibitor that does not cross the blood-brain barrier (BBB)[13]. Given that Trk receptor activity is essential for cognitive function[12], drugs incapable of crossing the BBB, such as GNF 5837 and PF 06273340[13], would have less cognitive side effects and thus might be more attractive for therapy; by contrast, ANA 12 does enter the central nervous system[14].

Therefore, we evaluated a range of doses of each drug (1.25 to 50 μM) in proliferating and senescent cells and established their EC50 (Fig. 2a). Analysis of viability and caspase 3/7 activity (an indicator of apoptosis) 48 h later revealed that the optimal senolytic doses were 10 μM GNF 5837, 20 μM ANA 12, and 30 μM PF 06273340 (Fig. 2b, Supplementary Fig. 2a), since they caused maximal death of senescent cells with minimal consequences on proliferating cells. Importantly, similar doses of TrkB inhibitors reduced the viability of several other primary cells that were subjected to etoposide-induced senescence (ETIS), including human BJ and IMR-90 fibroblasts, HUVECs (human umbilical vein endothelial cells), HSAECs (human small airway epithelial cells), and HRECs (human renal mixed epithelial cells). Similar reductions were seen in the viability of WI-38 cells that were rendered senescent by other means, specifically by exposure to ionizing radiation (IR) and by long-term culture until they reached replicative senescence (IRIS and RS, respectively) (Fig. 2c; Supplementary Fig. 2b–d). We confirmed the induction of senescence in these models by measuring SA-β-Gal activity and BrdU incorporation (Supplementary Fig. 2b–d). In this panel of 9 senescent populations, similar doses of TrkB inhibitors also significantly increased caspase 3/7 activity relative to untreated senescent cells (Fig. 2d). Importantly, these doses did not reduce the viability of proliferating cells (Supplementary Fig. 2e), supporting the notion that senescent cells selectively experienced increased apoptotic death in response to TrkB inhibition.

Finally, to confirm that TrkB inhibitors caused apoptotic cell death, we blocked caspase activity by incubating cells with the pan-caspase inhibitor Z-VAD-FMK along with GNF 5837, ANA 12, or PF 06273340. As shown, the increased cell death and caspase 3/7 activity seen by inhibiting TrkB were reversed in cells simultaneously treated with Z-VAD-FMK (Fig. 2e, f). These data support the notion that inhibiting Trk proteins enhanced pro-apoptotic signaling in senescent cells, leading to cell death.

### Elevated TrkB levels in senescent cells promotes resistance to apoptosis

We then set out to further characterize the role of Trk receptors in cellular senescence. In neuronal cells, activation of Trk receptors strongly promotes cell viability through specific gene expression programs[15]. Although our data pointed to a prominent pro-survival role for TrkB in senescent cells, we also analyzed the structurally similar TrkA and TrkC. Western blot analysis revealed that the levels of TrkA, TrkB [mostly full-length TrkB (FL) and not truncated TrkB (T)], as well as senescence marker p21 were low in proliferating (P) WI-38 fibroblasts, but increased robustly in WI-38 fibroblasts rendered senescent (S), as described in Fig. 2c (ETIS, OSIS, IRIS, RS) (Fig. 3a). As anticipated, we also found increased levels of *p16/CDKN2A* and *IL6* mRNAs, two markers of senescence, as quantified by reverse transcription (RT) followed by real-time quantitative (q)PCR analysis (Fig. 3b). We similarly assessed the levels of *NTRK1*, *NTRK2*, and *NTRK3* mRNAs (encoding TrkA, TrkB, and TrkC, respectively) by RT-qPCR analysis, but were unable to detect *NTRK3* mRNA in WI-38 fibroblasts,

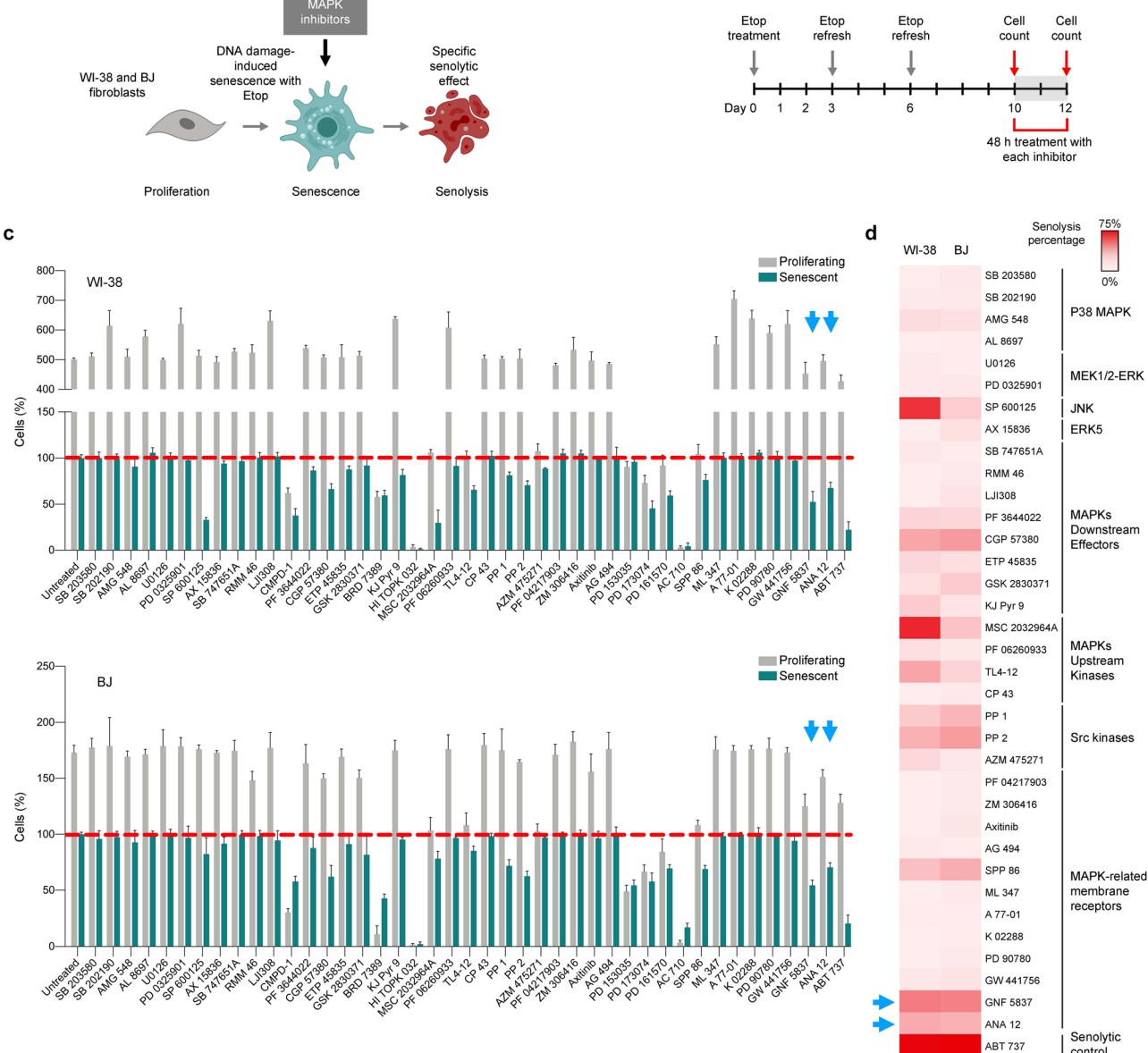

**Fig. 1 | Screening of a library of drugs directed at MAPK signaling proteins unveils Trk inhibitors as potential senolytic compounds. a** Overview of the experimental strategy of the survey. Briefly, senescent and proliferating WI-38 and BJ fibroblasts were treated for 48 h with a collection of MAPK inhibitors, whereupon cell viability was assessed. **b** Schematic representation of the protocol used to reach full senescence using etoposide (Etop, 25 and 50 µM for BJ and WI-38 cells, respectively) for 10 days, refreshing etoposide twice, before treating the cell cultures with each of the drugs in the library for an additional 48 h; created using BioRender. **c** Bar plot displaying the % change of proliferating and senescent WI-38 fibroblasts (top graph) and BJ fibroblasts (bottom graph) after treatment with the drug library for 48 h (day 12) relative to the time before drug addition (day 10). Red line, percentages of cells present at each condition right before starting the treatments at day 10. All the drugs were added at a concentration of 10 µM. **d** Heat map representing the percentage of senolysis (percentage of cell death) after 48 h when comparing day 12 to day 10 in senescent cells. Drugs were grouped by the target proteins inhibited within the MAPK superfamily. Graphs in **c** represent the mean values ±SD of $n = 3$ experiments. See also Supplementary Fig. 1.

and only *NTRK2* mRNA was consistently detected in all WI-38 S populations, and was generally higher than in P populations (Supplementary Fig. 3a). Similarly, the levels of TrkB protein, and *p16*, *IL6*, and *NTRK2* mRNAs were preferentially elevated in senescent BJ, IMR-90, HSAEC, HREC, and HUVEC cultures (Fig. 3c, d; Supplementary Fig. 3a).

To determine the role of each Trk receptor in senescent cell viability, we reduced the levels of TrkA or TrkB by transfecting specific small interfering (si)RNAs. After silencing was achieved, we triggered senescence with etoposide and monitored cell viability. As observed in Fig. 3e, neither cells in which TrkA was silenced (siNTRK1) nor control cells showed reduced viability, while silencing TrkB (siNTRK2) significantly decreased the number of live cells. The efficiency of the

silencing interventions was confirmed by measuring mRNA levels (Fig. 3f). In sum, despite the rise in TrkA levels in some senescence models, the enhanced viability of senescent cells appeared to be mediated by TrkB.

Next, since TrkB protein levels increased proportionally more than *NTRK2* mRNA levels in senescent cells, we evaluated the stability of TrkB protein by treating proliferating and senescent cells with cycloheximide (CHX), an inhibitor of translation. This analysis included positive control p53 (TP53), a protein showing increased stability in senescence[16,17]. Western blot analysis (Fig. 3g) revealed that TrkB was more stable in senescent fibroblasts, as was p53, suggesting that the increased TrkB protein stability contributes to its accumulation in

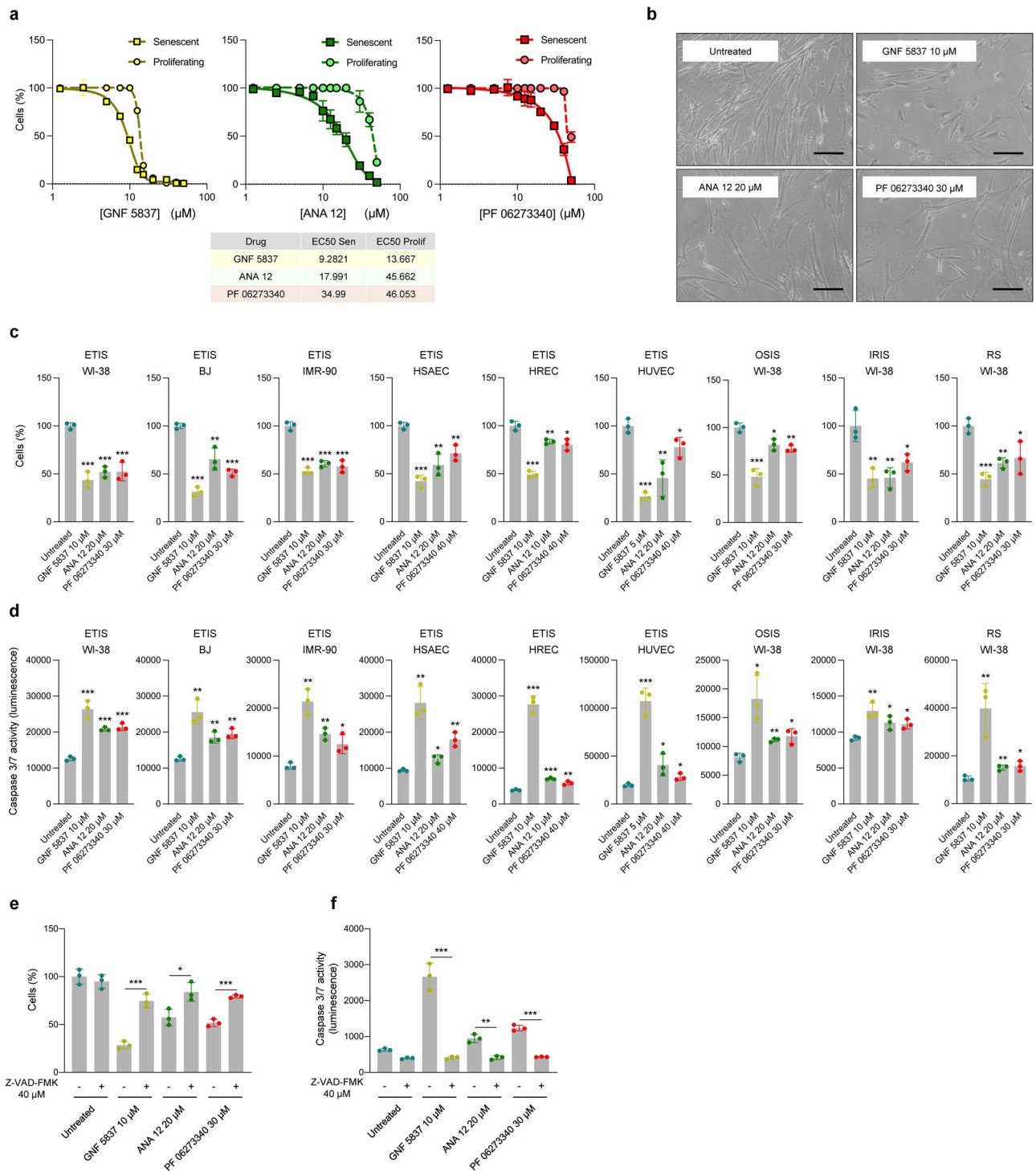

**Fig. 2 | Pharmacological inhibition of TrkB selectively induces apoptotic death of senescent cells. a** Viability curves obtained after testing a range of concentrations of Trk inhibitor drugs (GNF 5837, ANA 12, PF 06273340) for 48 h in WI-38 fibroblasts that were either proliferating or undergoing etoposide-induced senescence (ETIS). **b** Representative micrographs of select doses for each drug tested in (**a**); bar, 100 μm. Cell viability (**c**) and caspase 3/7 activity (**d**) were assessed 48 h after the indicated treatments and doses in WI-38, BJ, IMR-90, HSAEC, HREC, and HUVEC cultures undergoing ETIS, and WI-38 cultures undergoing oxidative stress-induced senescence (OSIS), ionizing radiation-induced senescence (IRIS), and replicative senescence (RS). Assessments of cell viability (**e**) and caspase 3/7 activity (**f**) measured in WI-38 cells undergoing ETIS that were treated for 48 h with different Trk inhibitors, either alone or in combination with the pan-caspase inhibitor Z-VAD-FMK at the indicated doses. Graphs in **a**, **c–f**, represent the mean values ± SD of $n = 3$ experiments; significance (*$p < 0.05$, **$p < 0.01$, ***$p < 0.001$) was determined by using two-tailed Student's $t$-test.

senescent cells. A time-course analysis after triggering senescence with etoposide revealed that TrkB protein levels increased by day 2 and continued to rise until day 10 (Fig. 3h). Early markers of senescence, p21 protein and *TGFB1* mRNA (Fig. 3h, i)[18–20], as well as late markers of senescence, *p16* and *IL6* mRNAs (Fig. 3i), were included in the analysis.

Although with different kinetics, the levels of TrkB as well as *TGFB1*, *p16*, and *IL6* mRNAs also increased progressively with senescence in BJ and IMR-90 cells (Supplementary Fig. 3b–e).

Finally, to study if TrkB was found on the plasma membrane of senescent cells, we enriched the proteins present on the plasma

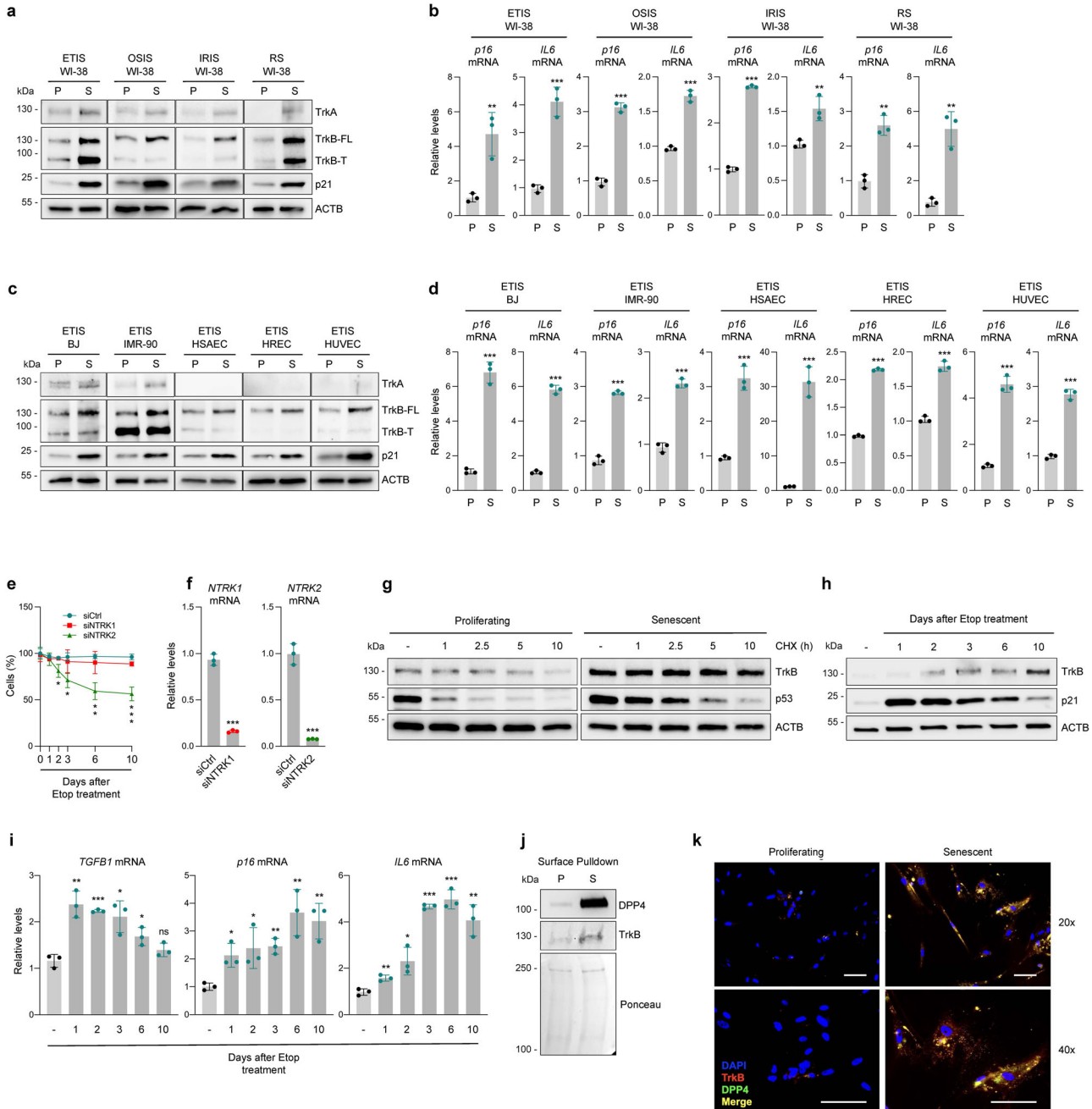

**Fig. 3 | Increased TrkB levels in senescent cells promotes resistance to apoptosis. a, b** In WI-38 fibroblasts that were proliferating (P) or undergoing senescence (S) (ETIS, OSIS, IRIS, RS as in Fig. 2c), western blot analysis was used to evaluate TrkA, TrkB full-length (FL), and truncated (T), p21, and ACTB (**a**), and RT-qPCR analysis to measure the levels of *p16* and *IL6* mRNAs (**b**). Western blot analysis (**c**) and RT-qPCR analysis (**d**) of the same molecules as in **a, b** but measured in BJ and IMR-90 fibroblasts and in HSAECs, HRECs, and HUVECs rendered senescent by ETIS (S) as described in Fig. 2c or proliferating (P). **e** WI-38 cells were transfected with siCtrl, siNTRK1 (directed at TrkA), or siNTRK2 (directed at TrkB) and treated with etoposide (ETIS); % live cells relative to day 0 were counted at the indicated times. **f** RT-qPCR analysis of the levels of *NTRK1* and *NTRK2* mRNAs at day 10 in the groups analyzed in **e**. **g** Western blot analysis of the levels of TrkB, labile protein p53, and ACTB in proliferating and senescent (ETIS) WI-38 cells at the times shown after treating with CHX (50 μg/ml). **h** Western blot analysis of the levels of TrkB, p21, and ACTB throughout senescence induction with etoposide in WI-38 cells. **i** RT-qPCR analysis of the levels of *TGFB1*, *p16*, and *IL6* mRNAs in the same conditions analyzed in **h**. **j** After pulldown of biotinylated cell-surface proteins from WI-38 cells [proliferating (P) or subjected to ETIS (S)], TrkB and DPP4 were assessed by western blot analysis. Ponceau staining of the transferred samples served to assess differences in sample loading and transfer. **k** Immunofluorescence micrographs showing cells positive for TrkB (red), DPP4 (green), and merged signals (orange/yellow) in the groups studied in **j** in non-permeabilizing conditions. Blue, DAPI staining to identify nuclei. Scale bar, 100 μm. In **a, c, g, h**, ACTB was included as loading control. Data in **b, d, f, i** were normalized to *ACTB* mRNA. Graphs in **b, d–f, i** display the mean values ± SD of *n* = 3 experiments; significance (*p < 0.05, **p < 0.01, ***p < 0.001) was determined by using two-tailed Student's *t*-test.

membrane using a biotinylation strategy (Methods) and found TrkB in the enriched 'Pulldown' sample; we included as a positive control DPP4, a protein reported to increase on the plasma membrane of senescent cells[21] (Fig. 3j). Detection of TrkB by immunofluorescence using non-permeabilizing conditions confirmed that TrkB was strongly abundant on the membrane of senescent cells, as was DPP4 (Fig. 3k). Together, these data indicate that TrkB protein increases during senescence in several cell systems, and contributes to ensuring senescent cell viability.

**BDNF is a SASP factor that ensures viability in senescent cells**

Trk receptors are activated by a family of secreted ligands known as neurotrophins, comprising NGF, NTF3, NTF4, and BDNF[15], which are essential for neuronal function. Even though BDNF is the preferred ligand of TrkB, we explored possible roles for all neurotrophins in this paradigm of cell senescence. First, by RT-qPCR analysis, we measured the levels of *NGF*, *NTF3*, *NTF4*, and *BDNF* mRNAs in all nine senescence models; interestingly, only *BDNF* mRNA was consistently elevated in all of them (Fig. 4a). Second, we found increased *BDNF* mRNA levels in published transcriptomic datasets[22] obtained from different models of senescence: in WI-38 fibroblasts rendered senescent by treatment with doxorubicin and in HUVECs and human aortic endothelial cells (HAECs) rendered senescent by treatment with IR (Supplementary Fig. 4a)[22]. Third, to test if BDNF is secreted by senescent cells, we collected media from the senescence models used in Fig. 4a; as shown, the levels of secreted BDNF, as analyzed by ELISA, were significantly higher in media collected from all senescent cells compared to the proliferating counterparts (Fig. 4b); in proliferating or senescent WI-38 cells, NGF, NTF3, and NTF4 were undetectable by ELISA (Supplementary Fig. 4b). The levels of BDNF and other SASP cytokines [CXCL1 (GRO-alpha), IL6, HGF, or CXCL10] were also elevated in media from senescent cells as detected by using a cytokine array although NTF3 and NTF4 were not (Fig. 4c). Together, these data indicate that a broad range of senescent cells secrete BDNF.

By immunofluorescence microscopy, we found that senescent WI-38 cells were strongly positive for BDNF (Fig. 4d), as more of the cells were positive (Fig. 4e) and the signals were more intense (Fig. 4f) than those observed in proliferating controls. RT-qPCR analysis likewise revealed an early rise in *BDNF* mRNA levels two days after triggering senescence by treatment of WI-38, BJ, and IMR-90 fibroblasts with etoposide (Fig. 4g). Since many SASP members are regulated by NF-κB and/or p53, we investigated if these two transcription factors induced BDNF production in senescent cells. Following p53 and RELA silencing by transfection of specific siRNAs (siTP53, siRELA), we observed that the rise in *BDNF* mRNA levels in senescent cells was dramatically suppressed by silencing p53 and modestly reduced by silencing RELA, compared with control (siCtrl) transfections (Fig. 4h and Supplementary Fig. 4c). The senescence-associated rise in *p21* mRNA levels, which is dependent on the transcriptional activity of p53, was similarly reduced by silencing p53, while induction of the senescence-associated *IL6* mRNA, an NF-κB-regulated mRNA, was mostly abrogated by silencing RELA. The rise in *p16* mRNA was not reversed by either silencing intervention (Fig. 4h).

We then asked if BDNF might directly affect senescent cell survival through TrkB activation, as it promotes neuronal viability[15,23]. We silenced BDNF by using a specific siRNA (siBDNF), and then we triggered senescence in WI-38 fibroblasts with either etoposide, H₂O₂, or IR, as well as in HUVECs with etoposide. As shown in Fig. 4i, silencing BDNF significantly reduced cell viability relative to control (siCtrl) cells throughout the process of reaching senescence; RT-qPCR analysis confirmed the persistence of silencing by 10 days (Supplementary Fig. 4d). Importantly, the loss of viability after BDNF silencing was partially rescued by supplementation of exogenous BDNF (200 pg/ml) (Fig. 4j). To further assess if the effect of BDNF was autocrine/paracrine, we employed anti-BDNF blocking antibodies (or non-specific IgG antibodies in control incubations); 48 h after adding the antibodies (at 4 µg/ml) to the media, proliferating WI-38 cells showed no change in replication or viability, while senescent WI-38 cells showed markedly decreased viability (Fig. 4k, l) and increased caspase 3/7 activity (Fig. 4m) in the presence of anti-BDNF antibodies.

Finally, BDNF cleavage by proteases such as Furin or matrix metalloproteinases (MMPs), which increase with senescence, is key to its function as a ligand of TrkB[24,25]. Western blot analysis of BDNF revealed overall higher levels of total BDNF and processed BDNF in conditioned media from senescent than proliferating WI-38 cells

(Fig. 4n). We tested if senescence-associated increases in Furin or MMPs might cause BDNF cleavage and activation, by first analyzing the levels of *FURIN*, *MMP1*, *MMP3*, *MMP7*, and *MMP9* mRNAs after etoposide treatment in WI-38, BJ, and IMR-90 cells; as shown, the levels of *FURIN* and *MMP3* mRNAs increased significantly during senescence (Supplementary Fig. S4e), while *MMP1* mRNA levels did not change, and *MMP7* and *MMP9* mRNAs were not detected in these cell types. We then tested if FURIN and MMP3 participated in BDNF processing during senescence by using Furin Inhibitor II (FURINi) and MMP Inhibitor II (MMPi). As shown, only treatment with MMPi reduced WI-38 cell viability and BDNF processing (Fig. 4o, p) during senescence, suggesting that MMP3 contributes to BDNF maturation during senescence. In sum, our results identify BDNF as a SASP factor whose abundance and function are regulated at several levels, and is implicated in promoting senescent cell survival in an autocrine and/or paracrine fashion.

**ERK5 activation by TrkB-BDNF axis sustains senescent cell survival through BCL2L2**

We then sought to investigate if the activation of TrkB by BDNF was linked to the TrkB-mediated survival of senescent WI-38 cells. First, to visualize protein phosphorylation we employed phos-tag polyacrylamide gels, which capture phosphorylated proteins and markedly reduce their migration, while non-phosphorylated proteins migrate at the expected size. As shown, slow-migrating TrkB bands appeared by 2 to 3 days of etoposide treatment, increasing in intensity in subsequent days (Fig. 5a, *left*). As the rise in phosphorylation mirrored the rise in *BDNF* mRNA abundance, we sought evidence that BDNF expression was linked to TrkB phosphorylation. As shown, silencing BDNF in WI-38 cells before triggering senescence completely prevented the appearance of the phosphorylated TrkB band (Fig. 5a, ***right***), suggesting that BDNF production is critical for inducing TrkB phosphorylation in senescent cells. Second, we tested whether both TrkB and BDNF depletion affected senescent cell viability in a similar time frame. Given that TrkB and BDNF levels rose markedly by 2 days into senescence, and that silencing either TrkB or BDNF increased cell death, we evaluated whether their depletion affected senescent cell survival at early and/or late senescence, two stages defined previously[19,26]. We found that silencing TrkB or BDNF decreased cell survival at both early and late senescence, although this effect was more pronounced in late senescence (Fig. 5b).

We then sequenced the bulk RNA present in late senescent cells (8 days into senescence) in which either TrkB or BDNF were silenced, and compared it to siCtrl-transfected senescent counterparts (Fig. 5c). The RNA-seq data are deposited in GSE202951. GSEA analysis of mRNAs differentially abundant after silencing TrkB or BDNF revealed enriched pathways of apoptosis and caspase activity (Fig. 5d, ***left***). The heat map in Fig. 5d, comprising differentially abundant mRNAs encoding apoptosis-related proteins, revealed jointly modulated transcripts including *BCL2L2* mRNA, which encodes a protein essential for the survival of senescent cells[8,27]. We used RT-qPCR analysis to validate these results by measuring the levels of *BCL2L2* mRNA, along with the levels of other mRNAs encoding senescence-associated proteins implicated in apoptosis, such as BCL2L1 and PUMA[6,8] (Supplementary Fig. 5a). Notably, while the levels of p53-induced pro-apoptotic *PUMA* mRNA were not affected, the levels of *BCL2L2* mRNA, encoding BCL2L2, decreased after TrkB or BDNF silencing at both early and late senescence (Fig. 5e). This effect appeared specific for *BCL2L2* mRNA, as *BCL2L1* mRNA levels were unchanged after silencing TrkB or BDNF (Fig. 5e), as were the levels of the p53-regulated *p21* mRNA, while *IL6* mRNA levels unexpectedly declined after silencing TrkB or BDNF (Supplementary Fig. 5a). Furthermore, BDNF-blocking antibodies specifically blunted the rise in *BCL2L2* mRNA in senescent cells, while *PUMA* and *BCL2L1* mRNAs were unaffected (Fig. 5f). As seen after silencing TrkB or BDNF in senescent cells (Supplementary Fig. 5a),

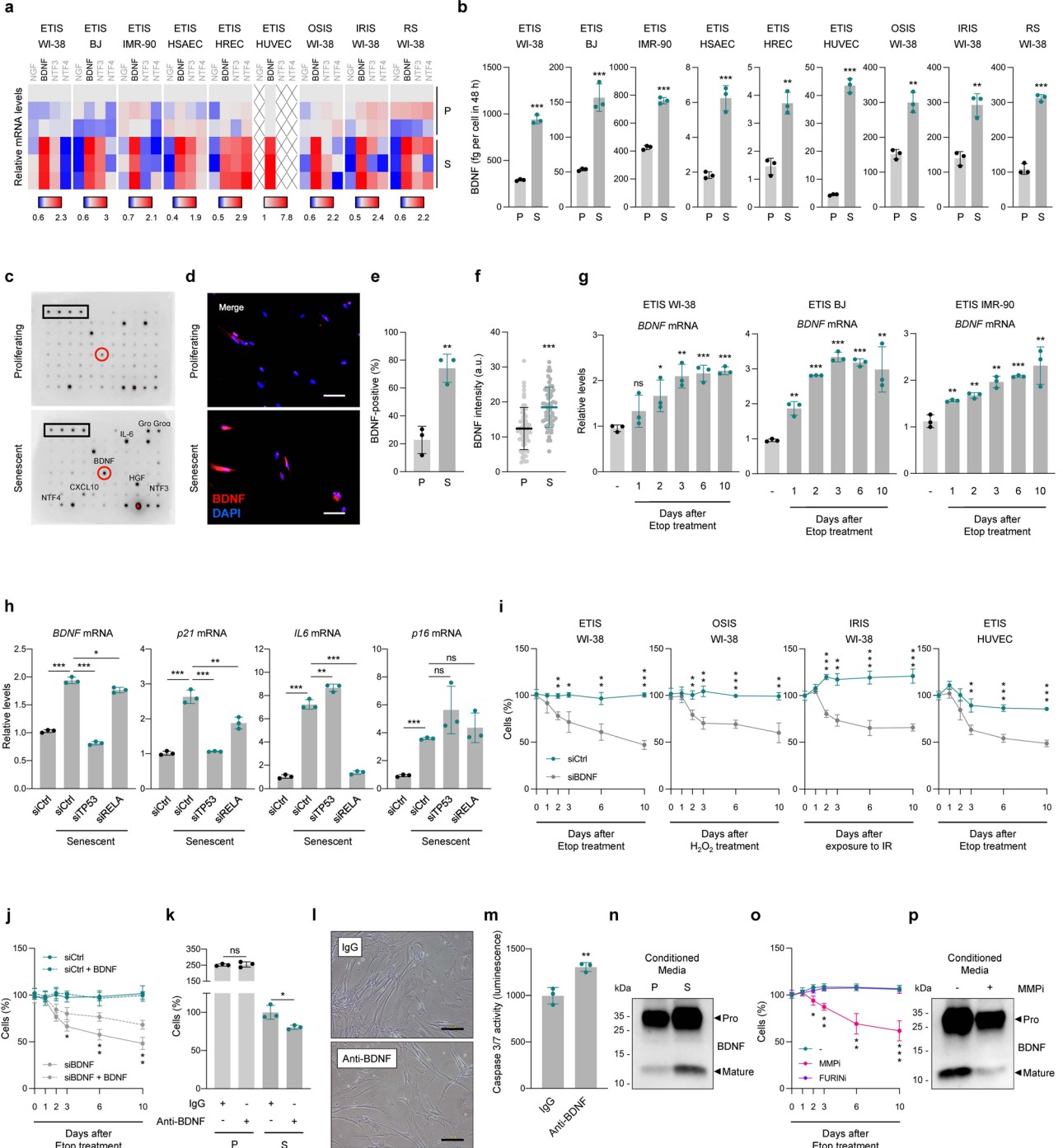

**Fig. 4 | BDNF is a SASP factor that ensures viability of senescent cells. a** Heat map of the levels of mRNAs encoding neurotrophins as quantified by RT-qPCR analysis in the senescence models described in Fig. 2c. **b** ELISA measurement of BDNF levels in conditioned media collected for 48 h in the groups shown in **a**. **c** Cytokine array analysis of SASP factors and neurotrophins in proliferating and senescent (ETIS) WI-38 fibroblasts. Immunofluorescence micrographs of BDNF signal (red) and DAPI-stained nuclei (blue) (**d**), % cells with positive BDNF staining (**e**), and BDNF fluorescence intensity measurements (**f**) in senescent (ETIS) WI-38 cells. **g** RT-qPCR analysis of *BDNF* mRNA levels in senescent (ETIS) WI-38, BJ, and IMR-90 fibroblasts. **h** RT-qPCR analysis of the levels of *BDNF*, *p21*, *IL6*, and *p16* mRNAs in senescent (ETIS) WI-38 fibroblasts transfected with the indicated siRNAs, relative to proliferating cells transfected with siCtrl. **i** WI-38 fibroblasts and HUVECs transfected with siCtrl or siBDNF were exposed to senescence-inducing treatments (ETIS, OSIS, IRIS); % cells remaining at each time point. **j** WI-38 fibroblasts were

transfected with siCtrl or siBDNF siRNAs, subjected to ETIS, and supplemented or not with exogenous BDNF (200 pg/ml, refreshed every 48 h); % cells remaining are shown. Proliferating and senescent (ETIS) WI-38 fibroblasts were treated with IgG or BDNF-blocking antibodies (4 µg/ml each); % remaining cells 48 h later were counted (**k**) and S cells were assessed by microscopy (**l**) and by caspase 3/7 activity (**m**). **n** Western blot analysis of BDNF levels and sizes in conditioned media from P and S (ETIS) WI-38 cells. **o** WI-38 cells undergoing ETIS were additionally treated with DMSO (−), MMP inhibitor II (MMPi, 10 µM), or Furin inhibitor II (FURINi, 15 µM). Remaining cells at the indicated times are shown. **p** Western blot analysis as in **n**, but additionally treated with DMSO (−) or MMPi during senescence induction. Scale bars, 100 µm. Data in **g**, **h** were normalized to *ACTB* mRNA. Graphs in **b**, **e**−**k**, **m**, and **o** represent the mean values ± SD of $n = 3$ experiments; significance (*$p < 0.05$, **$p < 0.01$, ***$p < 0.001$) was determined by using two-tailed Student's *t*-test.

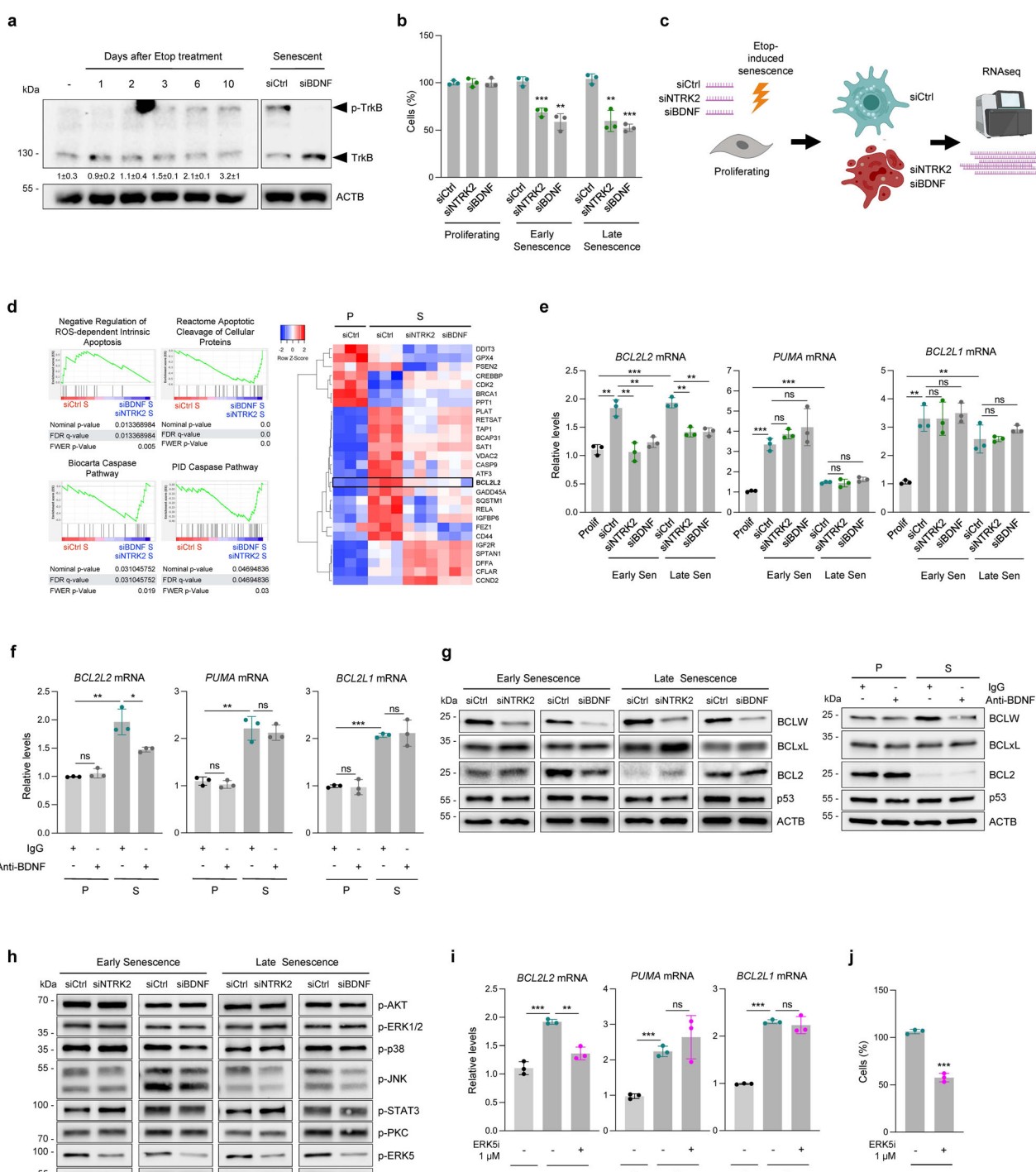

**Fig. 5 | ERK5 activation by TrkB-BDNF sustains senescent cell survival through BCL2L2. a** Western blot analysis using PhosTag gels to separate phosphorylated (p-TrkB) from unphosphorylated TrkB in WI-38 cells progressing to senescence (*left*), as well as in cells transfected to express normal (siCtrl) or reduced (siBDNF) BDNF levels (*right*). Phosphorylation ratios (p-TrkB/TrkB) were calculated (means ± SD) relative to day 0. **b** WI-38 fibroblasts transfected with siCtrl, siNTRK2, or siBDNF were treated with etoposide (50 μM) and cell viability was measured by direct cell counting at early senescence (day 2) and late senescence (day 8). **c** Experimental design for RNA-seq analysis to evaluate transcriptomic differences among cell groups; created using BioRender. **d** Transcriptomic analysis of WI-38 cells in the groups explained in **c**; GSEA associations (*left*), and heat map depicting genes related to apoptosis that changed significantly in cells transfected with siNTRK2 or siBDNF (*right*). **e** WI-38 cells were processed as in **b** and the levels of *PUMA*, *BCL2L2*, and *BCL2L1* mRNAs were quantified by RT-qPCR analysis. **f** WI-38 cells were treated with IgG or BDNF-blocking antibodies

(4 μg/ml each) for 48 h, and the levels of *PUMA*, *BCL2L2*, and *BCL2L1* mRNAs were quantified by RT-qPCR analysis. **g** WI-38 cells were processed as explained in **b**, **f** and the levels of the proteins shown were assessed by western blot analysis. **h** WI-38 cells were treated as explained in **b** and western blot analysis was used to assess the levels of effector proteins downstream of BDNF-TrkB, including p-AKT(S473), p-ERK1/2(T202/Y204), p-p38(T180/Y182), p-JNK(T183/Y185), p-STAT3(Y705), p-PKCα/β II(T638/641), and p-ERK5(T218/Y220). **i** RT-qPCR analysis of the levels of *BCL2L2*, *PUMA*, and *BCL2L1* mRNAs in proliferating (P) or etoposide-induced senescent (S) WI-38 cells treated or not with an ERK5 inhibitor (ERK5i, ERK5-IN-1; 48 h, 1 μM). **j** Cell viability as measured by direct cell counting of the remaining viable cells in the senescent groups described in **i**. In **a**, **g**, **h**, ACTB was included as loading control. Data in **e**, **f**, **i** were normalized to *ACTB* mRNA. Graphs (**b**, **e**, **f**, **i**, and **j**), represent the values ± SD from n = 3 experiments; significance (*p < 0.05, **p < 0.01, ***p < 0.001) was determined by using two-tailed Student's t-test.

treatment with BDNF-blocking antibodies did not change *p21* mRNA levels but reduced *IL6* mRNA levels (Supplementary Fig. 5b). Western blot analysis of BCL2L2 (BCLW) expression levels confirmed the changes in *BCL2L2* mRNA levels and further showed that p53 levels were unchanged (Fig. 5g); in agreement with these findings, all three TrkB inhibitors studied specifically reduced BCLW protein levels (Supplementary Fig. 5c).

Next, we analyzed the major pathways downstream of BDNF and TrkB signaling. Phosphorylation of ERK5 was dramatically reduced by silencing TrkB or BDNF, in both early and late senescence (days 2 and 8, respectively; Fig. 5h), a notable finding, given that ERK5 elevates BCL2L2 levels downstream of BDNF-TrkB for the survival of neuronal cells[28]. We then utilized a highly selective ERK5 inhibitor (ERK5-IN-1, 1 μM) to reduce ERK5 activity in senescent cells and found that it lowered both BCL2L2 levels and senescent-cell viability, but it did not affect proliferating cells (Fig. 5i, j and Supplementary Fig. 5d, e). In keeping with the results after silencing TrkB or BDNF (Supplementary Fig. 5a), inhibiting ERK5 specifically lowered the levels of *BCL2L2* and *IL6* mRNAs, but not the levels of *PUMA*, *BCL2L1*, or *p21* mRNAs (Fig. 5i and Supplementary Fig. 5f), while TrkB inhibitors reduced ERK5 phosphorylation levels (Supplementary Fig. 5g). In sum, our data suggest that activation of TrkB by BDNF specifically activates ERK5, which in turn increases BCL2L2 levels.

## BDNF is a discrete marker of survival in cellular senescence

In light of our earlier findings that p53 increases BDNF production, we investigated if BDNF induction contributed to the actions of p53 in implementing senescence or apoptosis programs. First, we treated WI-38 cells with a range of doses of etoposide that had different effects on cell proliferation and viability[11]. Untreated cells proliferated as expected, but increasing doses of etoposide reduced proliferation to different degrees (5 to 50 μM etoposide) or caused cell death (100 to 200 μM etoposide) (Fig. 6a). *BDNF* mRNA levels increased at etoposide concentrations that reduced proliferation (≤50 μM) but not at concentrations that triggered cell death (100–200 μM) (Fig. 6b); by contrast, *p21* and *IL6* mRNAs accumulated with increasing DNA damage (Fig. 6b). Increasing $H_2O_2$ doses similarly elevated *BDNF* mRNA levels only at sublethal levels (Fig. 6c, d), suggesting that other senescence-causing stressors function similarly in controlling *BDNF* mRNA production. Second, we modulated p53 function to study if p53 influenced *BDNF* mRNA levels. Silencing p53 prevented the rise in *BDNF* mRNA elicited by 50 μM etoposide treatment for 48 h (Fig. 6e) while it promoted apoptosis over senescence (Supplementary Fig. 2d); as anticipated, the same intervention reduced the levels of *p21* mRNA but not *IL6* mRNA (Supplementary Fig. 6a, b). These experiments underscore a requirement for functional p53 to induce BDNF and promote survival at early stages of senescence, despite the absence of p53 sites on the *BDNF* promoter. We then tested if increasing p53 function affected the production of BDNF after treatment with etoposide (50 μM for 48 h) by employing Nutlin-3a (Nut3a), a p53-stabilizing compound[29]. Notably, increasing p53 levels in cells treated with 50 μM etoposide significantly decreased BDNF levels (Fig. 6e), resembling the reduction observed at higher levels of DNA damage; in contrast, *p21* mRNA levels increased in the presence of Nut3a (Supplementary Fig. 6c) and Nut3a reduced cell viability (Supplementary Fig. 6d). These data suggest that, while p53 is required for BDNF induction after sublethal damage, only moderate p53 levels induce BDNF production and higher levels of p53 prevent the rise in BDNF that promotes survival.

Several mechanisms can explain how p53 could both induce and repress BDNF production depending on the extent of DNA damage. STAT3 was recently reported to promote BDNF expression in the lung upon injury[30] and we found that STAT3 phosphorylation at Tyrosine 705 (Tyr705) decreased in IMR-90 fibroblasts committed to apoptosis but remained phosphorylated in cells committed to senescence[11]. Therefore, we silenced STAT3 by transfection with siRNA

(Supplementary Fig. 6e) and evaluated BDNF levels after treatment with etoposide (50 μM, 48 h). As shown in Fig. 6f, the rise in *BDNF* mRNA triggered by senescence-inducing etoposide (50 μM) was prevented by STAT3 silencing; by contrast, *p21* mRNA levels were unchanged and the slight rise in *IL6* mRNA levels was suppressed (Supplementary Fig. 6e). Even though silencing STAT3 lowered BDNF levels, silencing STAT3 did not reduce viability, suggesting that STAT3 might be one of several factors governing the survival-apoptosis balance in early senescence (Supplementary Fig. 6f). Interestingly, both silencing and activating p53 decreased STAT3 phosphorylation (Fig. 6g), mirroring the effects seen for BDNF. Moreover, p53 levels increased linearly along with DNA damage (Fig. 6h), but STAT3 phosphorylation at Tyr705 peaked at 25 μM, when p53 levels were moderately elevated. Similar responses were seen when comparing $H_2O_2$ doses causing senescence or apoptosis (Fig. 6i), indicating that this mechanism of cell fate determination is not limited to etoposide.

We then performed single-cell RNA-seq analysis to evaluate if different subpopulations of senescent WI-38 cells expressed different *BDNF* mRNA levels, and if cells expressing higher BDNF levels displayed a transcriptomic profile associated with STAT3 activation. After clustering cells into different subgroups (Supplementary Fig. 6g), *BDNF* mRNA levels were highest in clusters 4 and 5 and lowest in clusters 0 and 1 (Fig. 6j, Supplementary Fig. 6h). *BDNF* mRNA levels correlated significantly with mRNAs transcriptionally upregulated by STAT3, such as *THBS1* or *FN1* mRNAs, as determined by Gene Set Enrichment Analysis (GSEA; Fig. 6j, k, and Supplementary Fig. 6i) and by ChIP-seq-identified targets (Supplementary Fig. 6h)[31]. Clusters 4 and 5 were linked to Epithelial-Mesenchymal Transition (EMT) programs and low DNA damage levels (Supplementary Fig. 6j), in agreement with cell programs associated with senescence and survival[11]. In validation experiments, immunofluorescence analysis revealed that most cells expressing the highest levels of BDNF were also strongly positive for THBS1 or FN1 (Supplementary Fig. 6k), two proteins encoded by STAT3-regulated mRNAs (cluster 5, Fig. 6j).

Finally, we studied if both p-STAT3 and BDNF were simultaneously present at a single-cell level during senescence in WI-38 fibroblasts and in an in vivo mouse model of doxorubicin-induced senescence (Supplementary Fig. 6l). By double immunofluorescence staining of senescent WI-38 cells with antibodies recognizing p-STAT3(Tyr705) and BDNF, we found that most BDNF-positive cells were p-STAT3(Tyr705)-positive, but virtually every p-STAT3(Tyr705)-positive cell was BDNF-positive (Fig. 6l, m). Similarly, in the mouse model, lung cells positive for the DNA damage marker γH2AX were also positive for p-STAT3 (Tyr705) and BDNF (Supplementary Fig. 6m). These results indicate that p53 influence on STAT3 activation can both induce and repress BDNF production depending on the level of DNA damage. In turn, BDNF secretion by senescent cells can trigger signaling through TrkB → ERK5 → BCL2L2 to ensure survival in an autocrine manner (Fig. 6n).

## Delivery of TrkB inhibitors reduces senescent cell burden in old mice

Finally, we tested the relevance of the BDNF-TrkB paradigm on senescent cell viability in vivo using C57BL/6 J mice. We chose naturally aged mice as a well-established and relevant in vivo cellular senescence model[32]. The appearance of senescent cells in older tissues such as lung and liver has been documented by different methods, such as assessment of SA-β-Gal activity and detection of senescence markers like p16[33–35]. Initially, we studied the extent to which p16- and BDNF-positive cells colocalized in tissues in 24-month-old (m.o.) mice; we employed an antibody recently shown to recognize p16[34] and verified the detection of p16-positive cells by seeing colocalization of tdTomato from a knock-in mouse strain that expressed tdTomato from the endogenous *p16Ink4* gene[36] (Supplementary Fig. 7a, b). Approximately one-half of the p16-positive cells in lung and liver were also BDNF-positive (Fig. 7a), suggesting that the two proteins were co-expressed in aging.

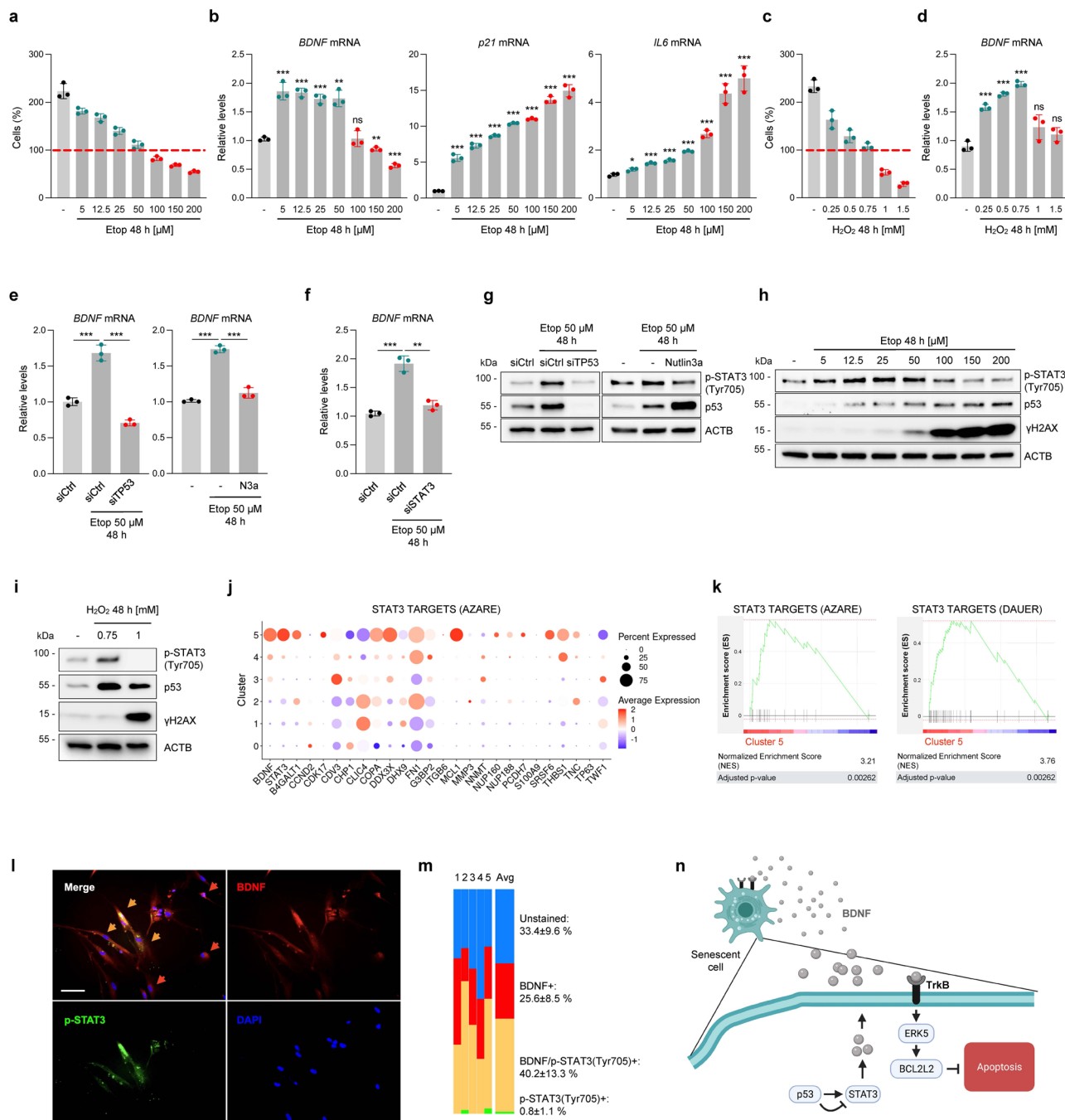

**Fig. 6 | BDNF is a marker of surviving senescent cells.** WI-38 fibroblasts were treated with the doses of etoposide (Etop) shown; 48 h later, cell viability was assessed by direct cell counts and represented relative to initial counts (red line) (**a**) and RT-qPCR analysis was used to measure the levels of *BDNF*, *p21*, and *IL6* mRNAs (**b**). WI-38 cells were treated with $H_2O_2$ and viability (**c**) was measured as in **a** and *BDNF* mRNA levels (**d**) were measured as in **b**. **e** RT-qPCR analysis of *BDNF* mRNA levels in WI-38 cells treated with etoposide (50 μM, 48 h) after silencing p53 (*left*) by transfection with siTP53 (siCtrl in control transfections) or inducing p53 levels (*right*) by treatment with Nut3a (N3a, 10 μM). **f** RT-qPCR analysis of *BDNF* mRNA levels in WI-38 cells transfected with either siCtrl or siSTAT3 siRNAs and treated for 48 h with 50 μM etoposide. Western blot analysis of p-STAT3(Y705) and p53 protein levels in WI-38 cells treated as in **e** (**g**); p-STAT3(Y705), γH2AX(S139), and p53 protein levels in the groups described in **a** (**h**), and in $H_2O_2$-treated WI-38 cells as described in **c** (**i**). **j** Expression levels of STAT3-regulated mRNAs in GSEA gene set

'STAT3 TARGETS AZARE' across the different clusters set in single-cell RNA-seq analysis of senescent (50 μM etoposide, 8 days) relative to proliferating WI-38 cells. Dot size and color represent the percentage of cells expressing a transcript and the average expression value, respectively. **k** GSEA plots displaying enrichment scores of gene sets 'STAT3 TARGETS AZARE' and 'STAT3 TARGETS DAUER' for cluster 5 from the analysis in **j**. **l** Immunofluorescence analysis of colocalized signals for p-STAT3(Y705) (green) and BDNF (red) in senescent WI-38 cells (50 μM etoposide, 8 days). Blue, nuclei stained with DAPI; orange arrows, cells co-stained for p-STAT3 and BDNF; red arrows, BDNF-only positive cells. Scale bar, 100 μm. **m** Quantification of signals from **l**; percentages ±SD of the resulting staining for each group described. **n** Schematic depicting the proposed model described in this study; created using BioRender. Data in **b**, **d**, **e**, **f** were normalized to *ACTB* mRNA. Values in **a–f** are the means ± SD; significance (*$p < 0.05$, **$p < 0.01$, ***$p < 0.001$) was determined by using two-tailed Student's *t*-test.

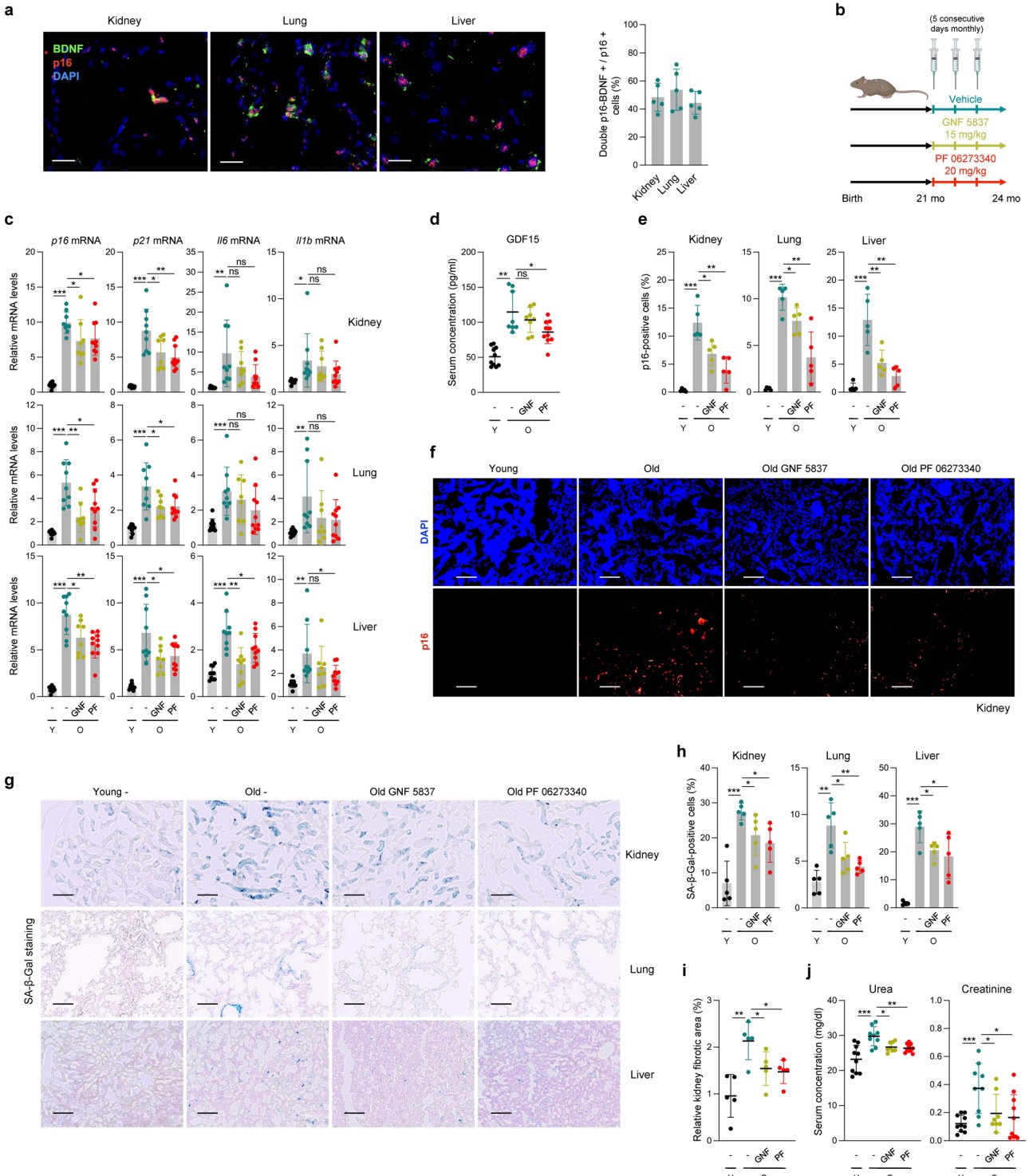

**Fig. 7 | Treatment with TrkB inhibitors reduces senescent cells in organs from old mice. a** Immunofluorescence staining of BDNF (green) and p16 (red) in kidney, lung, and liver from 24-month old (m.o.) C57BL/6 J mice. Scale bars, 20 μm. The percentages of double p16- and BDNF-positive cells among the total p16-positive cells were analyzed (*n* = 5 mice, 24 m.o.) and plotted (*right*). **b** Schedule of administration of TrkB inhibitors GNF 5837 and PF 06273340 to normally aging mice. Groups included are Y (young, 3 m.o.) and O (old, 24 m.o.); - (untreated), GNF (GNF 5837), and PF (PF 06273340). Created using BioRender. **c** RT-qPCR analysis of the abundance of *p16*, *p21*, *Il6*, and *Il1b* mRNAs in kidney, lung, and liver in the different mouse cohorts. **d** GDF15 concentration in sera from the four mouse cohorts, as measured with a Bioplex instrument. Percentages of p16-positive cells

in kidney, lung, and liver of the four mouse cohorts (**e**), and representative images of p16-positive cells in the kidney of the groups tested (**f**). Representative images (**g**) and quantification (**h**) of SA-β-Gal activity assessment performed in the same four mouse cohorts described in **b**. **i** Percent relative fibrotic area measured as the fraction of Sirius Red-positive area present in the images taken from *n* = 5 mice from each of the groups described in **b**. Additional data in Supplementary Fig. 7f. **j** Serum concentration of urea and creatinine measured in the mouse groups described in **b**. Scale bars in **f** and **g**, 200 μm. Data in **c** were normalized to *Actb* mRNA. Graphs in **c**–**e**, and **h**–**j** reflect individual data points; values are displayed as means ± SD; significance (*$p < 0.05$, **$p < 0.01$, ***$p < 0.001$) was determined by using two-tailed Student's *t*-test.

We then tested the ability of TrkB inhibitors to reduce cell senescence markers in old mice. We administered the Trk inhibitors characterized in Fig. 2 by monthly injections into 21 m.o. mice until they were 24 m.o. (Fig. 7b and Supplementary Methods). We assayed the same inhibitors we had tested in culture (GNF 5837, ANA 12, and PF 06273340), but halted treatments with ANA 12 due to toxicity; in fact, ANA 12 was a poor candidate for therapy due to its ability to penetrate the BBB[14]. Notably, compared with control mice [vehicle-treated 24 m.o. and untreated young (3 m.o.) mice], GNF 5837 and PF 06273340 showed promising effects on senescence markers and traits. The increases in levels of *p16* and *p21* mRNAs observed in kidney, lung, and liver of old mice were significantly mitigated by each drug (Fig. 7c); the same trends were seen for *Il6* and *Il1b* mRNAs, but they did not reach significance in all organs (Fig. 7c). Similarly, the levels of *Bdnf* mRNA and the senescence marker *Gdf15* mRNA were not significantly elevated in kidney, lung, and liver of old mice (Supplementary Fig. 7c), possibly because the percentages of senescent cells in organs are typically low, and/or senescent cells may not express *Bdnf* mRNA at all stages of senescence. These observations indicate that these drugs, which do not cross the BBB, were able to decrease the levels of prominent senescence markers such as *p16* or *p21* mRNAs in several tissues in aged mice.

We then studied senescence-related serum markers that were recently found to increase with age[37]. One such marker, GDF15, increased markedly in old mice; this increase was significantly mitigated by treatment with PF 06273340, while GNF 5837 showed the same trend without reaching significance (Fig. 7d). Other serum markers (TIMP-1 and PAI-1) were not elevated with age (Supplementary Fig. 7d) and thus they could not be used to test the effects of these drugs; importantly, these drugs did not affect body weight (Supplementary Fig. 7e). Immunofluorescence analysis further revealed that both GNF 5837 and PF 06273340 treatments reduced the percentage of p16-positive cells in all three organs tested (Fig. 7e and Supplementary Fig. 7f). As was the case for GDF15 serum levels, PF 06273340 appeared to be more effective than GNF 5837 in reducing p16 levels. Assessment of the canonical SA-β-Gal marker revealed that both drug treatments reduced the age-associated increases observed both in the intensity and the number of SA-β-Gal-positive cells in kidney, lung, and liver (Fig. 7g, h).

Finally, we asked if the observed reductions in senescent cells ameliorated physiological declines linked to cell senescence during aging. First, we assessed tissue fibrosis by monitoring extracellular matrix deposition using Sirius (Picro Sirius) red staining. As shown, old tissues displayed increased Sirius red staining that was significantly reduced with both TrkB inhibitors (Supplementary Fig. 7g). Furthermore, we quantified the area of fibrosis in the kidney in each group, and measured different serum markers that were increased with an age-associated loss of renal fitness, such as serum urea and creatinine levels; all three markers of renal disfunction increased with age but the increase was ameliorated by treatment with either GNF 5837 or PF 06273340 (Fig. 7i, j). Finally, given earlier reports that these drugs do not cross the blood-brain barrier[13,38], we asked if these drugs affected structure or cell viability in the brain cortex. As shown, there was no sign of toxicity as determined by terminal deoxynucleotidyl transferase dUTP nick end labeling (TUNEL) analysis to measure apoptosis, staining for neuronal density using the neuronal marker NeuN, or measuring the total number of cells by staining nuclei using DAPI (Supplementary Fig. 7h, i). Together, we propose that the reduction in senescence markers is evidence that treatments with GNF 5837 and PF 06273340 represent promising strategies to reduce the accumulation of senescent cells with aging.

## Discussion

Strategies to remove detrimental senescent cells from tissues and organs are gaining interest in the clinic, particularly for the treatment of chronic age-related diseases. Given that senescent cells bring about a proinflammatory state through the SASP, they have been found to exacerbate many aging-associated pathologies[4,39]. Therefore, extensive efforts are underway to develop pharmacological approaches to eliminate them with minimum side effects[5,40]. Here, the discovery that Trk inhibitors selectively reduced senescent cell viability (Fig. 1) led us to propose a previously unrecognized role for TrkB in supporting the survival of senescent cells. Although a role for TrkB in senescence was not previously described, TrkB is a well-established pro-survival receptor in neurons[10,12,15] and thus we set out to characterize the function of TrkB in senescent cell survival.

Interestingly, the levels of TrkB ligand BDNF, a neurotrophin, were found specifically elevated in several senescence models (Fig. 4a, b), and interfering with BDNF function by silencing or blocking antibodies caused a specific reduction in the viability of senescent cells (Figs. 4, 5). The active secretion of molecules by senescent cells through the SASP is a key feature of the response to damage, and SASP factors help to establish a repair program that includes the promotion of fibrosis, inflammation, recruitment of immune cells, and remodeling of the tissue architecture. The discovery that the SASP factor BDNF promotes senescent cell survival by activating TrkB uncovers an important dimension of the SASP in strengthening the anti-apoptotic attributes of senescent cells.

The finding that a neuronal survival paradigm is utilized by senescent cells to support viability was interesting, but it was not entirely unexpected, as activation of TrkB by BDNF (and more generally Trk receptors by neurotrophins) has been described outside of the nervous system, particularly in developmental and tissue repair processes in several organs[30,41-43]. On the other hand, aberrant activation of the BDNF-TrkB axis has been described in age-related pathologies such as lung fibrosis, sarcopenia, cancer, and kidney disease[44-48]; interestingly, all these processes causing age-associated declines and diseases have been linked to cellular senescence[49-54]. Thus, it will be important to determine if BDNF secreted by senescent cells contributes to these paradigms by influencing surrounding cells, and possibly by promoting an EMT-like program[46,55].

In cells facing damage that might result in either senescence or death, production of BDNF appears to be a better predictor of a path to senescence than other, more established, senescence markers such as p21 or IL6 (Fig. 6), which are also strongly induced by acute DNA damage but are not always good indicators of senescence[56-58]. Furthermore, given that depleting BDNF levels in cells undergoing senescence drastically reduced viability (Figs. 4, 5), BDNF appears to function as a key effector of the anti-apoptotic program of senescence. In sum, the activation of BDNF-TrkB in senescent cells supports the notion that senescence is a developmental program set in motion by cell damage that shares previously unrecognized molecular similarities with other cell responses such as neuronal survival. In fact, the activation of TrkB by BDNF enhanced ERK5 signaling to increase the levels of BCL2L2, a trait shared with neurons[28]. Together with the fact that BCL2L2 was identified as a core senescence-associated protein elevated in multiple cell senescence models[27], and the fact that the senolytic drug ABT-737, an inhibitor of BCL2L2, potently sensitizes senescent cells to apoptotic death[7,8], a prominent role for BCL2L2 in senescent cell survival is becoming increasingly apparent. In further support for this notion, we recently found that BCL2L2, but not BCL2L1, contributed to sustaining cell viability in early senescence in a model of senescence that relied on the activation of the oncoprotein SRC[11]. Moreover, while direct inhibition of pro-survival BCL2 proteins can cause notable side effects[59], reductions in BCL2L2 by suppressing BDNF-TrkB signaling might offer advantages over other therapies.

A major caveat that must be overcome before the BDNF-TrkB paradigm can be exploited therapeutically is the fact that BDNF-TrkB activity is required to maintain a fully functional nervous system[60]. Since Trk inhibition is used to treat chronic pain[61], various compounds

have been developed that prevent adverse side effects within the central nervous system. Two out of the three drugs tested in this study, GNF 5837 and PF 06273340, are peripherally restricted Trk inhibitors, i.e., they cannot cross the blood-brain barrier[13,38], and thus avoid impacting neuronal function. Nonetheless, these compounds reduced senescence-associated markers in tissues such as kidney, lung, and liver in a mouse model of senescence triggered by old age (Fig. 7 and Supplementary Fig. 7). Eventually, it will be essential to study if these or other TrkB inhibitors are viable strategies to reduce the burden of senescent cells in human conditions of dysfunction and disease. In sum, we have identified a role for BDNF in promoting cell survival through the activation of TrkB, and propose that this signaling paradigm might be exploited therapeutically to lower senescent cells in older organs.

## Methods

### Cell culture and treatment

Human IMR-90 (ATCC), WI-38 (Coriell Institute), and BJ fibroblasts (ATCC) were cultured in Dulbecco's modified Eagle's medium (DMEM, Gibco) supplemented with 10% heat-inactivated fetal bovine serum (FBS, Gibco), 0.5% Penn/Strep (Gibco), Sodium Pyruvate (Gibco), and non-essential amino acids (Gibco) in a 5% $CO_2$ incubator. Human umbilical vein endothelial cells (HUVECs), small airway epithelial cells (HSAECs), and renal mixed epithelial cells (HRECs), all from ATCC, were cultured in their respective media [vascular cell basal medium plus endothelial cell growth kit-BBE (bovine brain extract), airway epithelial cell basal medium plus bronchial epithelial cell growth kit, and renal epithelial cell basal medium plus renal epithelial cell growth kit, respectively], supplemented with 0.5% Penn/Strep (Gibco), Sodium Pyruvate (Gibco) and non-essential amino acids (Gibco), and cultured under the same conditions (5% $CO_2$ incubator). Cells were maintained at low population doubling levels (PDL) for the different experiments included in this article. Cellular senescence was triggered by different means. Etoposide-induced senescent was achieved by culturing WI-38, BJ, and IMR-90 fibroblasts for 10 days in the presence of etoposide (Selleckchem) at 50, 25, and 50 μM, respectively. To achieve senescence by exposure to ionizing (γ) radiation (IR), WI-38 cells were exposed to 15 Gray (Gy) and cells were cultured for up to 10 days. Replicative senescence was achieved after serial passaging of WI-38 cells until replicative exhaustion (typically at ~PDL55). Oxidative stress-induced senescence was achieved by adding 0.75 mM $H_2O_2$ directly to WI-38 fibroblasts in complete medium and replacing fresh medium 2 h later; cells were then assayed at the indicated times. Etoposide-induced senescence was achieved by treating HUVECs (at 10 μM), HSAECs (at 20 μM), and HRECs (at 20 μM) for 3 days, whereupon media was refreshed without etoposide until senescence was reached at day 8. All drugs and compounds used were refreshed every 48 h. The concentration of exogenous BDNF (R&D Systems) was calculated based on the observed levels of BDNF production by ELISA (200 pg/ml, refreshed every 48 h). The doses of MMP and FURIN inhibitors (MMP inhibitor II and FURIN inhibitor II, Sigma-Aldrich), Nutlin3a (Selleckchem), Z-VAD-FMK (Selleckchem), and the library of MAPK inhibitors (Tocris) are indicated throughout the manuscript. The drugs used are listed (**Source Data**).

Cells were transfected with RNAiMAX (Invitrogen) following the manufacturer's instructions. Briefly, cells at 50% confluency were transfected with ON-TARGETplus SMARTPool (Dharmacon) non-targeting siCtrl (Catalog ID: D-001810-10-05), siNTrk1 (Catalog ID: L-003159-00-0005), siNTrk2 (Catalog ID: L-003160-00-0005), siTP53 (Catalog ID: L-003329-00-0005), siRELA (Catalog ID: L-003533-00-0005), siBDNF (Catalog ID: L-017626-00-0005), or siSTAT3 (Catalog ID: L-003544-00-0005) siRNAs at a final concentration of 25 nM; 24 h later, treatments were initiated as indicated. Cell viability was assessed by direct cell counting; all cell counts were performed manually by using ImageJ and performed in at least 3 independent replicates. From each replicate, 3 fields were randomly selected and counted. Cell viability was represented as the percentage of remaining cells compared to the number of cells present at the beginning of the experiment.

**RT-qPCR analysis.** Tissues or cells were lysed in either Tri-Reagent (Invitrogen) or RLT buffer (Qiagen), and the lysate was processed with the QIAcube (Qiagen) to purify total RNA, which was then reverse-transcribed (RT) to create cDNA using Maxima reverse transcriptase (Thermo Fisher Scientific) and random hexamers. Real-time, quantitative (q)PCR analysis was then performed using SYBR Green mix (Kapa Biosystems), and the relative expression was determined by the $2^{-\Delta\Delta Ct}$ method. The levels of mRNAs were normalized to human *ACTB* mRNA or mouse *Actb* mRNA levels.

The primers used for human transcripts, each forward (F) and reverse (R) were: GTTACGGTCGGAGGCCG and GTGAGAGTGGCGGGGTC for *p16/CDKN2A* mRNA; AGTCAGTTCCTTGTGGAGCC and CATGGGTTCTGACGGACAT for *p21/CDKN1A* mRNA; AGTGAGGAACAAGCCAGAGC and GTCAGGGGTGGTTATTGCAT for *IL6* mRNA; CTTCCAGCCGAGGTCCTT and CCCTGGACACCAACTATTGC for *TGFB1* mRNA; GATGGTGGCCTACCTGGAGA and AGAGCTGTGAACTCCGCCA for *BCL2L2* mRNA; GGCTTGACATCATTGGCTGAC and CATTGGGCCGAACTTTCTGGT for *BDNF* mRNA; GCAAGGCTGATAACGCTGAGGA and CCTGGGCATCAGCGGTCAATG for *NTF4* mRNA; CAAGCAGATGGTGGACGTTAAGG and TCGCAGCAGTTCGGTGTCCATT for *NTF3* mRNA; ACCCGCAACATTACTGTGGACC and GACCTCGAAGTCCAGATCCTGA for *NGF* mRNA; CCGACACTGTGGTCATTGGCAT and CAGTTCTCGCTTCAGCACGATG for *NTRK3* mRNA; TCGTGGCATTTCCGAGATTGG and TCGTCAGTTTGTTTCGGGTAAA for *NTRK2* mRNA; CACTAACAGCACATCTGGAGACC and TGAGCACAAGGAGCAGCGTAGA for *NTRK1* mRNA; CATGTACGTTGCTATCCAGGC and CTCCTTAATGTCACGCACGAT for *ACTB* mRNA; GACCTCAACGCACAGTACGAG and AGGAGTCCCATGATGAGATTGT for *PUMA* mRNA; GAGCTGGTGGTTGACTTTCTC and TCCATCTCCGATTCAGTCCCT for *BCL2L1* mRNA; CCTCAGCATCTTATCCGAGTGG and TGGATGGTGGTACAGTCAGAGC for *TP53* mRNA; ATGTGGAGATCATTGAGCAGC and CCTGGTCCTGTGTAGCCATT for *RELA* mRNA; TCGGGGACTATTACCACTTCTG and CCAGCCACTGTACTTGAGGC for *FURIN* mRNA; CGGTTCCGCCTGTCTCAAG and CGCCAAAAGTGCCTGTCTT for *MMP3* mRNA; and CAGCAGCTTGACACACGGTA and AAACACCAAAGTGGCATGTGA for *STAT3* mRNA.

The primers used for mouse transcripts, each forward (F) and reverse (R) were: CTGCAAGAGACTTCCATCCAG and AGTGGTATAGACAGGTCTGTTGG for *Il6* mRNA; GAAATGCCACCTTTTGACAGTG and TGGATGCTCTCATCAGGACAG for *Il1b* mRNA; CCCAACGCCCCGAACT and GCAGAAGAGCTGCTACGTGAA for *p16/Cdkn2a* mRNA; TTCTTTGCAGCTCCTTCGTT and ATGGAGGGGAATACAGCCC for *Actb* mRNA; and TTGCCAGCAGAATAAAAGGTG and TTTGCTCCTGTGCGGAAC for *p21/Cdkn1a* mRNA.

**Caspase 3/7 activity.** Apoptosis-related caspase 3/7 activity was assayed by using the caspase-Glo® 3/7 Assay System (Promega). Briefly, caspase-Glo® 3/7 solution was added directly to each well. The plate was then shaken vigorously for 30 s and incubated at 25 °C in the dark for 30 to 180 min. Luminescence was measured using a GloMax plate reader (Promega) and normalized to cell numbers.

**Immunofluorescence analysis in cultured cells.** Cells were fixed in 4% PFA (paraformaldehyde) for 10 min and then permeabilized by incubating cells with 0.2% TX-100 for 5 min at 25 °C, unless stated otherwise. After blocking with 10% goat serum for 1 h, primary antibodies were added in the same 10% goat serum buffer and incubated overnight at 4 °C. Antibodies recognized phosphorylated STAT3 (Tyr705, 1:50, Cell Signaling Technology 9145), BDNF (1:40, Santa Cruz Biotechnology sc-65514), TrkB (1:40, Santa Cruz Biotechnology sc-

377218), DPP4 (1:100, Cell Signaling Technology 67138), FN1 (1:50, Abcam ab2413), and THBS1 (1:50, Abcam ab85762).

Fluorescent signals were detected by adding fluorescent secondary antibodies [Invitrogen, Goat anti-Mouse IgG (H + L) Cross-Adsorbed Secondary Antibody, Alexa Fluor 568; and Goat anti-Rabbit IgG (H + L), Superclonal Recombinant Secondary Antibody, Alexa Fluor 488] in 10% goat serum buffer (1:1000) for 1 h at 25 °C. Finally, DAPI was used to counterstain the nuclei; images were taken using a fluorescence microscope (BZ-X Analyzer, Keyence) and analyzed with ImageJ. The intensity of BDNF immunofluorescence was calculated using ImageJ by analyzing the fluorescent signal present within an area of 10 μm radius from the center of the DAPI-stained nuclei. A total of 60 cells were analyzed per experimental group (20 from each replicate).

**Western blot analysis.** Protein extracts were obtained by lysing cells with a denaturing buffer containing 2% sodium dodecyl sulfate (SDS) (Sigma-Aldrich) in 50 mM HEPES. After boiling and sonication, whole-cell protein extracts were size-separated through polyacrylamide gels and transferred to nitrocellulose membranes (Bio-Rad). Membranes were blocked with 5% non-fat dry milk and immunoblotted. Specific primary antibodies were used that recognized phosphorylated p38 MAPK (T180/Y182, Biolegend, Ref. 903501), phosphorylated SAPK/JNK (T183/Y185, 81E11, Cell Signaling Technology, Ref. 4668S), phosphorylated ERK1/2 (T202/Y204, Biolegend, Ref. 675502), phosphorylated AKT (Ser473, Cell Signaling Technology, Ref. 4060S), phosphorylated STAT3 (Y705, D3A7, XP®, Cell Signaling Technology, Ref. 9145S), p53 (DO-1, Cell Signaling Technology, Ref. 18032S), ACTB (β-Actin C4, Santa Cruz Biotechnology, sc-47778), and BCL2L2/BCL-w (Cell Signaling Technology, Ref. 2724S), phosphorylated PKCα/β II (Thr638/641, Cell Signaling Technology, Ref. 9375 S), phosphorylated ERK5 (Thr218/Tyr220, Cell Signaling Technology, Ref. 3375), BCLxL (54H6, Cell Signaling Technology, Ref. 2764S), BCL2 (D55G8, Cell Signaling Technology, Ref. 4223S), phosphorylated Histone H2A.X (Ser139, 20E3, Cell Signaling Technology, Ref. 9718S), p21 (Santa Cruz Biotechnology, sc-53870), TrkA (Cell Signaling Technology, Ref. 2510S), and TrkB (Santa Cruz Biotechnology, sc-136990). After incubation with the required secondary antibodies conjugated with horseradish peroxidase (HRP, Jackson Immunoresearch), the chemiluminescent signals were detected by using the Chemidoc system (Bio-Rad). For western blot analysis of conditioned media, the protein present in 1 ml of medium used for 48 h was precipitated by adding 250 μl of 6.1 N trichloroacetic acid (TCA, Sigma-Aldrich). The mixture was left on ice for 30 min, centrifuged at 14,000 × g for 30 min, washed 3 times in acetone, dried out, and resuspended in 100 μl of 2% SDS and 50 mM HEPES. The processing of BDNF was analyzed by western blot analysis using an anti-BDNF antibody (EPR1292, Abcam, ab108319). To assess TrkB phosphorylation, SuperSep™ Phos-tag™ Precast Gels (Fujifilm) were employed, as they markedly slow down the migration of phosphorylated proteins while unphosphorylated proteins migrate at the expected size. The densitometry analysis of western blots was carried out with ImageJ.

**Isolation of cell surface proteins.** Cell surface proteins were isolated by using the Pierce™ Cell Surface Biotinylation and Isolation Kit (ThermoFisher Scientific, Catalog No. A44390). Briefly, cells were washed twice with PBS and cell surface proteins were labeled for 10 min at 25 °C with a solution of EZ-Link Sulfo-NHS-SS-Biotin, a membrane-impermeable biotinylation reagent, followed by several washes in TBS. Cells were then lysed with the detergent provided in the kit and the same amount of protein extract between different samples was used for the subsequent pulldown. The biotinylated proteins were captured by biotin-affinity purification with a NeutrAvidin™ Agarose Resin (Product No. 29200, ThermoFisher Scientific), followed by several washes with the wash buffer. Dithiothreitol (DTT) (10 mM) was used in the elution buffer to reduce the disulfide bonds in the biotin label and release the bound proteins.

**Mice.** All mouse work, including the import, housing, experimental procedures, and euthanasia, were approved by the Animal Care and Use Committee (ACUC) of National Institute on Aging (NIA). C57BL/6 J Mice (50% male, 50% female) were provided standard chow *ad libitum* and maintained under a 12:12 h light/dark cycle. For the delivery of Trk inhibitors, 21-month-old mice were injected intraperitoneally for three months and euthanized at 24 months of age. The treatments were performed at the beginning of each month for 5 consecutive days, at the indicated doses (GNF 5837 at 15 mg/kg, PF 06273340 at 20 mg/kg).

**Bioplex analysis of mouse serum.** Mouse plasma was collected, allowed to clot for 2 h at 25 °C, and centrifuged for 20 min at 2000 × g. Serum was removed and frozen at −80 °C. For the multiplex assay, serum was thawed and centrifuged at 16,000 × g for 4 min. A custom murine Luminex Assay kit was designed by R&D Biosystems to include the following analytes: GDF-15, PAI-1, and TIMP-1. Serum was diluted 1:2 using Calibrator Diluent RD6-52 provided in the kit. Standards (provided with the kit), blanks and serum were incubated with the microparticle cocktail for 2 h at 25 °C, followed by incubation with a biotin-antibody cocktail for 1 h. A final incubation of 30 min with Streptavidin-PE was performed with shaking at 25 °C before running the plate on the Biorad Bioplex-200 instrument. Each incubation was followed by three sets of washes with wash buffer (provided in the kit). Instrument settings were adjusted to the following: 50 μl sample volume, Bio-Plex MagPlex Beads (Magnetic), Double Discriminator Gates set at 8000 and 23,000, low RP1 target value for the CAL2 setting, 50 count/region. The results were analyzed with the Bio-Plex Manager software.

**BrdU incorporation.** Cells were incubated with (4 μg/ml) 5-Bromo-2′-deoxyuridine (BrdU) diluted in DMEM media with 10% FBS for 24 h. BrdU incorporation was measured following the manufacturer's protocol (Cell Signaling Technology). Briefly, cells were fixed and denatured prior to the addition of BrdU mouse mAb, and BrdU was detected using the GloMax plate reader (Promega).

**Detection of SA-β-Gal.** Senescence-associated β-Galactosidase (SA-β-Gal) activity was assessed following the manufacturer's instructions (Cell Signaling Technology). For the in vivo samples, dried slides were pre-washed with PBS plus 1 mM Magnesium Chloride before the addition of staining solution. All the slides were stained at once for 16 h at 37 °C. The resulting images were captured by a fluorescence microscope (BZ-X Analyzer, Keyence) and analyzed with ImageJ.

**Tissue processing.** To perform RT-qPCR analysis of tissue samples, organ pieces were flash-frozen in liquid nitrogen and preserved at −80 °C until required. To extract RNA, tissue samples were introduced in Tri-Reagent (Invitrogen) and disrupted using a tissue homogenizer (Bertin Instruments). RNA extraction was then carried out as indicated by the manufacturer's instructions.

For histological analysis, tissue biopsies were immediately fixed in 4% PFA in PBS at 4 °C overnight. The following day, tissues were cryoprotected in 30% sucrose solution in PBS at 4 °C overnight, and subsequently included in OCT embedding compound, and stored at −80 °C until needed. For tissue cutting, OCT blocks were processed using a cryostat at −20 °C to obtain 10-μm sections that were mounted onto Superfrost™ Plus Microscope Slides (Fisher Scientific) and dried out overnight at 25 °C. Before incubating with antibodies, antigen retrieval was performed with a sodium citrate-based buffer (Abcam) following the manufacturer's instructions. Tissue slides were then permeabilized with 0.2% TX-100 in PBS for 10 min, blocked with goat serum blocking solution (Biolegend) for 1 h, and incubated overnight with the antibodies. The antibodies used for immunofluorescence

recognized BDNF (EPR1292, Abcam, ab108319; 1:100 dilution in blocking solution) and CDKN2A/p16INK4a (2D9A12, Abcam, ab54210; 1:250 dilution in blocking solution). Fluorescent secondary antibodies (Invitrogen) were added in 10% goat serum for 1 h at 25 °C and slides were mounted with ProLong Diamond Antifade Mountant with DAPI (Life Technologies) and coverslips.

Immunofluorescence analysis to detect multiple proteins was performed using antibodies that recognized γH2AX (ab140498, Abcam), p-STAT3 (Tyr705, 4113, Cell Signaling Technology), and BDNF in lung, as well as with antibodies that detect p16 (2D9A12, Abcam, ab54210) and tdTomato (TA180009, Thermo Fisher) in liver. Immunofluorescent detection of neuronal marker NeuN (24307, Cell Signaling Technology) together with TUNEL staining in brain cortex were performed by iHisto (iHisto.io). Staining with Sirius red (Picro Sirius Red Stain Kit, ab150681, Abcam) was performed following the manufacturer's instructions.

**Cytokine array analysis.** Media were collected from cultured cells and centrifuged for 10 min at 1000 × g to remove any precipitates. The arrays (Cytokine Array C5 from RayBiotech) containing the pre-adsorbed antibodies, were blocked for 1 h at room temperature and incubated overnight with the media samples. The arrays were then incubated with a detection antibody and with detection buffers, and developed and analyzed on a Bio-Rad ChemiDoc Imaging System. Arrays included reference points as negative and positive controls for relative quantification.

**ELISA measurement of neurotrophins.** The concentration of neurotrophins (BDNF, NGF, NT-3, and NT-4) in conditioned media was quantified by performing individual ELISA assays (Raybiotech) following the manufacturer's instructions. Conditioned media were collected after 48 h of culture in different cell conditions (proliferation or senescence). Subsequent analyses were performed without media dilution. Values were normalized to cell counts at the time of media collection.

**Urea and Creatinine measurement.** Urea and creatinine assay kits (MAK006 and MAK080, respectively, Sigma-Aldrich) were used with mouse serum following the manufacturer's protocol.

**RNA-seq analysis.** Bulk RNA was extracted with RLT buffer (Qiagen) and purified using the QIAcube system (Qiagen) using RNeasy plus. The quality and quantity of RNA were assessed using the Agilent RNA 6000 nano kit on the Agilent Bioanalyzer. High-quality RNA (125 ng) was used to prepare a sequencing library using Illumina TruSeq Stranded mRNA Library prep kit following the manufacturer's protocol (Illumina, San Diego, CA). The quality and quantity of the libraries were checked using the Agilent DNA 1000 Screen Tape on the Agilent Tapestation. Paired-end sequencing was performed for 103 cycles with an Illumina NovaSeq sequencer. BCL files were de-multiplexed and converted to FASTQ files using bcl2fastq program (v2.20.0.422). FASTQ files were trimmed for adapter sequences using Cutadapt version v1.18 and aligned to human genome hg19 Ensembl v82 using STAR software v2.4.0j. featureCounts (v1.6.4) software was used to generate gene counts. The Bioconductor package DESeq2 version 1.30.0[62] in R (version 4.0.3) was used to compare counts levels; statistical testing was carried out with Wald test. Transcripts were considered differentially regulated if absolute log2 fold change was >1 and the Benjamini–Hochberg adjusted *p*-value was <0.05. Functional analysis of the differentially expressed genes was performed by using normalized counts in the GSEA platform[63].

Single-cell RNA-seq libraries were prepared according to the user guide of the Chromium Next GEM Single Cell 3′ Reagent Kits v3.1 (10x Genomics). Briefly, etoposide-induced senescent WI-38 cells were trypsinized, washed with PBS, and resuspended in 10% FBS and 0.1 mM EDTA in PBS at a concentration of 900–1,000 cells/μl. The single-cell suspension was loaded into a Chromium Next GEM Chip G (10x Genomics), and GEMs were generated using the Chromium Single Cell Controller (10x Genomics). After 11 and 13 cycles of cDNA amplification and Sample Index PCR (Single Index Kit T Set A, 10x Genomics), paired-end sequencing was performed on a SP100 flow cell on an Illumina NovaSeq platform. The raw single-cell RNA-seq data were processed using Cell Ranger software 6.0.0 (10x Genomics) and sequencing reads were mapped to the pre-built human reference (GRCh38) (version 2020-A, 10x Genomics). Filtered matrix files generated by Cell Ranger were imported and analyzed using R package Seurat 4.1.0. Cells expressing <200 or >6,000 genes and with >10% expression of mitochondrial genes were excluded from downstream analysis. The data were normalized and linearly transformed, and were subjected to variable features identification, principal component analysis, cluster analysis, and dimensional reduction analysis (UMAP), following the standard pre-processing workflow for single-cell RNA-seq data in Seurat. Gene Set Enrichment Analysis (GSEA) was performed using R packages including Seurat 4.1.0, msigdbr 7.4.1, fgsea 1.18.0, and presto 1.0.0, with identification of 'HALLMARK_IL6_JAK_STAT3_SIGNALING' from hallmark gene sets, and 'DAUER_STAT3_TARGETS_UP' and 'AZARE_STAT3_TARGETS' from curated gene sets are enriched in BDNF-expressing single-cell clusters. RNA-seq data (bulk and single-cell) are deposited in GSE202951 (https://www.ncbi.nlm.nih.gov/geo/query/acc.cgi?acc=GSE202951).

**Schematics and statistical analysis.** The schematics in Figs. 1 and 5–7 were created using BioRender, with a license to the NIA IRP. Data are presented as the means ± standard deviation (S.D.) of at least *n* = 3 independent experiments. Individual data points are displayed in all the bar plots. Significance was established using two-tailed Student's *t*-test (*$p < 0.05$, **$p < 0.01$, ***$p < 0.001$). All analyses were performed with Prism 9.

**Reporting summary**
Further information on research design is available in the Nature Research Reporting Summary linked to this article.

## Data availability
The RNA-sequencing data generated in this study are deposited in the Gene Expression Omnibus (GEO) with the accession code GSE202951. All other data generated or analysed during this study are included in this published article (and its supplementary information files). Source data are provided with this paper.

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

## Acknowledgements
This work was supported in its entirety by the National Institute on Aging Intramural Research Program of the National Institutes of Health. We thank S. Camandola (NIA) for help with the mouse experiments.

## Funding

## Competing interests
The authors declare no competing interests.
