## [Peer Review File · Nature Communications]

Title: A BDNF-TrkB autocrine loop enhances senescent cell viabilityREVIEWER COMMENTS

Reviewer #1 (Remarks to the Author):

The manuscript submitted by Anerillas et al., reports that a BDNF/TrkB axis activation is involved in maintaining the survival of senescent cells, inhibited by TrkB inhibitors.

Authors use etoposide to trigger cell senescence while previous studies in cancer models (neuroblastoma, sarcoma) have demonstrated that the BDNF/TrkB signaling pathway is involved in resistance to etoposide-induced apoptosis. Hence, it is probably not surprising that following etoposide exposure, similar mechanism with TrkB signaling induced cell survival following BDNF secretion and TrkB binding. Therefore, a form of confusion may arise between actual cell senescence or tolerance/resistance to etoposide or doxorubicin. Likewise, cellular senescence models should mainly highlight the main results obtained by other senescence induction models.

The work carried out requires clarification, particularly from technical points. In this way, this study in its current version does not present a substantial conceptual advance in BDNF/TrkB signaling pathway allowing cell survival upon etoposide exposure despite its drug is used as senescence inducer. Why human normal fibroblasts were not studied in addition to cell lines? or fibroblasts from old mice? Several methodological points and results must be clarified and completed.

Questions and major remarks:

- Methods:

The authors claim that BDNF belongs to the SASP, but their demonstration is still performed on human fibroblast cell lines, the authors would moderate their conclusions.

Indeed, authors would use human normal fibroblast to conclude that BDNF belong to SASP. Likewise, authors have to use different compound inducing senescence to rule out etoposide (PMID: 27145455, PMID: 29959561 and PMID: 25700942) or doxorubicin (PMID: 12438277) dependent mechanisms.

Ionized-irradiation (Gy irradiation?) is not described in methods. What is the reason to use 25 or 50 etoposide in Wi38 and BJ cells respectively?

Many "DNA damage" were cited by authors in the results but how was evaluated DNA damage? Notably they have not shown its specific markers such γ -H2A-X and telomere-associated DNA damage in senescent fibroblasts.

SiRNA experiments have to be more detailed in particular for the time of SiRNA exposition upon etoposide. Indeed, it seems difficult to keep target extinction at 10 days.

- Fig 1 c and Supp table 1:

Experimental conditions (concentrations and IC50 evaluation for drugs testing are not indicated). Legend of supp Table 1 is missing.

Why Pan Trk inhibitor (GNF 5837) has superior effect to TrkB inhibitor (ANA 12) and TrkA inhibitor? Depending on its concentrations, it could interfere with PDGFR (cited by Tocris as following “Exhibits selectivity for Trk receptors over a range of kinases, with some activity at PDGFR and c-Kit (IC50 values are 0.87 and 0.91 μ M respectively)”. Concentrations are not indicated.

- Likewise, SRC kinase belongs to TrkB cell signaling, PP2 inhibitor presents a close profile to ANA12 by decreasing senescent cell %. These results remain not discussed in the current manuscript.

- Fig 2: Method used for evaluation of cell viability is not clear. Cell counts of senescent (beta-Gal) or proliferating (BrdU) cells?

Fig 2 h and i: what are the cells? WI-38?

- Western blots:

Fig 3: b and e: Western blots are not representative and whole blots are required with MW markers to define the full-length TrkB receptor (145 kDa) with tyrosine kinase domain and the two other truncated forms (95 kDa), TrkB-T1 and TrkB-shc without kinase domain, both detected by anti-TrkB Ab sc-136990 used in this study.

Same remark for Supp Fig 3 a and c.

As for Fig 5 a: MW western-blot of P-TrkB / TrkB are absent, the size differences are surprising.

Quantification of Western-blot will be performed to assess the reproducibility of results.

- Immunofluorescence staining: images are not convincing.

Fig 3i: immunofluorescence staining for DPP4 is not clearly detectable, notably in contrast to the study from Kim et al. (ref. 21) from the same group that showed that DPP4 was strongly labelled in WI-38 senescent cells. There are too few cells per field, TrkB labelling is not only detected on the membrane and is also detected in the cytoplasm (while the authors indicate that the cells have not been permeabilized, therefore, a cytoplasmic diffusion in dead cells cannot be eliminated). Isotype antibody controls are not shown. Why the authors did not perform confocal microscopy analyses like Kim et al (21)? In this way, Mander's coefficient would be better to interpret colocalization with DPP4.

- Fig 4:

a) The heat map shows that not only BDNF mRNA was detected but also NT3 mRNA in BJ cells (Fig 4 a) and in WI-38 RS (Supp. Fig 4 a). (Why authors have used WI-38-doxorubicin in this model, not described in Methods?) These data do not match when they are compared to those of ELISA (not detecting NT3) and array (c: detecting NT3). Furthermore, in senescent HUVECs, NT3 mRNA levels were higher than those of BDNF. These discrepancies suggest that NT3 is also produced during senescence depending on the cell models and technical methods and that other NTs could be induced during senescence induction. This point is not discussed.

b) The ELISA procedure for BDNF and other neurotrophins measurement is not indicated. Western blotting of the supernatant of senescent cells is required to determine the type of BDNF, mature (14 kDa) or immature proBDNF (32 kDa), both detected by the sc-65514 used by the authors. This point is important given that the functions of proBDNF differ from those of its mature form (with different

binding properties and coreceptor), this point requires further elucidation. Furthermore, it was not shown that the metalloproteinases associated with proBDNF cleavage (matrix metalloproteinases 3, 7...) in mature BDNF are expressed in senescent cells.

fig 4 k: The parameters obtained from the non-relevant antibody appear to be incorrect, underestimating labelling of basal BDNF in tissues.

-Fig 6: co-staining STAT3-BDNF was not clearly superimposable (Fig 6 g). So, what is the meaning of the sentence “virtually every p-STAT3 positive cell was BDNF positive”?

- Fig 7: Experiments with old mice showed that BDNF is expressed by few co-expressing p16. It is difficult to understand why authors have preferentially used anti-BDNF instead of TrkB for their colocalization with p16 in mice tissues. According that TrkB/BDNF is upregulated in senescent tissues, experiments with lung, Kidney and liver with TrkB and BDNF costaining is necessary to assess their involvements during organs aging in old control mice by comparison to Trk inhibitors-treated mice.

In addition, considering that senescent fibroblasts models were obtained with induced-fibroblast cell lines models, a lacking experiment would be used to validate this model with old mice fibroblasts from skin, i.e.: BDNF, TrkB and beta-Gal expressions in old control mice fibroblasts should be compared to those obtained in Trk inhibitors-treated mice.

In Discussion: Other reports have demonstrated that normal human senescent fibroblasts secrete BDNF (Tinaburri et al., J Invest Dermatology 2021), inducing EMT in keratinocytes through TrkB activation in a paracrine loop; this is important to be discussed in the interest of inhibiting BDNF/TrkB activation in the senescent fibroblasts. The discussion could also be extended to CAF, in many cancers.

Minor issues to be addressed:

- 1/ The line numbers are missing which makes reviewing quite difficult.
- 2/ The method for preparing culture supernatants for analysis of released cytokines is missing.
- 3/ Likewise, the kit number of cytokine assay remains missing in the materials and methods section.
- 4/ Methods to assess drug library also remains missing, how authors obtain percentage of cells? How this percentage is calculated?

Reviewer #2 (Remarks to the Author):

The manuscript by Anerillas et al, entitled “A BDNF-TrkB autocrine loop enhances senescent cell viability”, describes the identification of a BDNF-TrkB pathway that facilitates survival of at least certain types of senescence cells. The facts that TrkB and BDNF appear to be upregulated in senescence cells in

culture and in vivo to facilitate survival of the damage cells and that BDNF colocalizes with p16 expression, making it a potential senescent cell biomarker, are both of significant interest. Thus the manuscript has the potential to make a significant contribution to the field of senescence and aging. However, there are several minor comments which the authors should try to address either experimentally or in the text of the manuscript.

Specific Questions:

1. How specific are the TrkB inhibitors GNF5837 and ANA12? Do they have any off target effects or effects on other receptor tyrosine receptors?
2. Dasatinib, a “dirty” tyrosine kinase inhibitor, is a known senolytic. Is there any evidence that dasatinib inhibits TrkB?
3. Do the TrkB inhibitors work as senolytics on other senescent cell types such as HUVECs, HAECs, astrocytes, preadipocytes or immune cells?
4. The EC50s for the TrkB inhibitors should be provided. Also, it should be determined if the IC50 for TrkB kinase inhibition is the same as the EC50 for senolytic activity.
5. It is unclear what the authors define as early and late senescence. Also, Line-1 is a better marker of late senescence than p16, which is elevated in early senescence.
6. Why were 1 month time points chosen for treatment of old mice with the TrkB inhibitors? Other groups have dosed more frequently with senolytics since the burden of SnCs increases several weeks post senolytic treatment. How long after the last treatment were the mice sacrificed?
7. The results suggest that BDNF, if it indeed is a SASP factor, should increase in the plasma with aging. Also expression of BDNF should go up in tissues and cell types in vivo. The inclusion of at least some evidence for increased BDNF expression at the RNA level in tissues and at the protein level in serum should be included or at least mentioned.
8. Antibodies against murine p16INK4a are notoriously bad. The authors need to provide supplemental data that the antibody used for the in vivo analysis of senescent cells indeed recognizes p16 such as through the use of p16 deficient cells or mice.
9. Is TrkB, which appears to be upregulated at the protein level and present on the cell surface, a better portien marker than BDNF for senescent cells? What percent of p16+ cells in vivo are TrkB+.
10. There are numerous panels of senescence and SASP factors that have been reported including CellAge and Sen_Mayo. However, BDNF is not a component of these genes set, possibly because the increase in expression of BDNF in senescent cells is only 2 fold at best. The authors need to discuss why

certain senescent/SASP gene sets, used to identify and characterize senescent cells, haven't picked up BDNF. Also, how does a two fold increase in BDNF mRNA lead to such high expression of protein so that it allows for identification of senescent cells by IF?

11. The regulation of BDNF at different levels of DNA damage is of interest but does raise questions regarding how it is regulated by p53, STAT3 and additional pathways. Is a similar differential regulation observed with replicated senescence, senescence induced by inflammatory factors, oncogene-induced senescence and/or IR. Although a full examination of this question would be difficult, analysis of BDNF expression following a different inducer of senescence would be appropriate.

Reviewer #3 (Remarks to the Author):

In the current study, Anerillas et al. reported a novel Senescent Cell Anti-Apoptotic Pathways (SCAPs) that senescent cells survive through a BDNF-TrkB autocrine loop. They first screened a group of compounds affecting MAPK pathways and then discovered that inhibitors targeting the TrkB receptor had the maximum senolytic effects. They further demonstrated that senescent cells highly produce BDNF binding to TrkB to activate the ERK pathway, which upregulates known SCAPs, Bcl-W. The discovery of novel SCAPs is exciting and important, extending the therapeutic potentials of senolytics. However, I have several main concerns:

1. The authors used two types of fibroblast cells to test the BDNF-TrkB autocrine loop in this study. To test whether the BDNF-TrkB autocrine loop is cell type-dependent, can the authors assess this SCAP in other types of cells such as endothelial cells or mouse MEFs?

2. In the previous publication (PMID: PMID: 31945054), the senescence-associated secretory phenotype (SASP) of senescent fibroblast cells was fully characterized using a comprehensive proteomic approach. However, BDNF was not found highly secreted by senescent cells. The discrepancy between studies might be caused by the induction of senescence, X-irradiation, and inducible RAS overexpression in the previous study, and chemo-drug (etoposide) was used to induce senescence in the current study. It raises whether the BDNF-TrkB autocrine loop solely depends on the etoposide induction. Can the authors use X-irradiation or Oncogene induce senescence to test the BDNF-TrkB autocrine loop?

3. In Figures 1 and 2, the TrkB inhibitors showed the most effective senolytic activities among other compounds. I noticed that EC50 of TrkB inhibitors is 10 times lower than known senolytics, ABT-263/ABT-737. Does this suggest additional independent signaling contributing to senescent cell survival? If yes, the authors should discuss it in the discussion section.

4. The authors used the term "early senescence" and "late senescence" to distinguish the changes of Bcl-2 families during senescence in the cultured cells. Can the authors clarify the "early" and "late"

stages? Does early senescence exist in vivo? Or is this an observation of the pre-senescence stage? Maybe a time course study is more helpful.

5. In the experiment of knocking down BDNF to test senescent cell viability, it would be more convincing if the authors could show adding soluble BDNF to prevent senescent cell death.

6. It is known that the survival of senescent fibroblast relies on Bcl-xl, Bcl-2, and Bcl-w. In figure 5g, it would be clearer if the authors could show the protein expression of Bcl-2 and Bcl-xl after knocking down NTRK2 and BDNF.

7. The authors speculate that the p-STAT3 pathway primes the production of BDNF. If so, will Jak/STAT inhibitor induce senescent cell death after etoposide induction?

8. In in vivo study, the authors showed the reduction of senescence markers in multiple tissues. It would be more convincing if the authors could provide more evidence of improvement of age-related phenotypes in the treated animals. Also, the cognitive test is essential to show no detrimental effects on CNS.

9. In figure 7d, one SASP factor, GDF15, was shown decreased by TrKB inhibitors. What about other well-known SASP factors? This is very important because it may indicate that TrKB inhibitors only target a specific type of senescent cells.

Minor suggestions

1. In the figure legend of figure 1c, the concentration of compounds is missing.

2. I noticed that the authors used DPP4 as a marker of senescent cells in figure 3. p16 is the most used senescence marker. It would be more convincing if the authors could provide the protein expression of p16 in etoposide-induced senescent cells and show how the protein level of TrKB and BDNF changes along with p16.

3. In figure 4d, the graph to show the quantification of intensities of BDNF is missing.

4. In figure 7a, the antibody validation for the immunofluorescence staining of p16 in mouse long and liver is required.

5. A detailed description of statistical analysis is missing in both the figure legend and Method section.

Response to the comments from Reviewer 1:

The manuscript submitted by Anerillas et al., reports that a BDNF/TrkB axis activation is involved in maintaining the survival of senescent cells, inhibited by TrkB inhibitors.

Authors use etoposide to trigger cell senescence while previous studies in cancer models (neuroblastoma, sarcoma) have demonstrated that the BDNF/TrkB signaling pathway is involved in resistance to etoposide-induced apoptosis. Hence, it is probably not surprising that following etoposide exposure, similar mechanism with TrkB signaling induced cell survival following BDNF secretion and TrkB binding. Therefore, a form of confusion may arise between actual cell senescence or tolerance/resistance to etoposide or doxorubicin. Likewise, cellular senescence models should mainly highlight the main results obtained by other senescence induction models.

The work carried out requires clarification, particularly from technical points. In this way, this study in its current version does not present a substantial conceptual advance in BDNF/TrkB signaling pathway allowing cell survival upon etoposide exposure despite its drug is used as senescence inducer. Why human normal fibroblasts were not studied in addition to cell lines? or fibroblasts from old mice? Several methodological points and results must be clarified and completed.

[AU] We thank the reviewer for these opening comments. We should begin with an apology for not stating more clearly in our original text that we did not use any cell lines (transformed or immortalized). Instead, we only used only primary human diploid fibroblasts: WI-38 and IMR-90 lung fibroblasts and BJ skin fibroblasts. These fibroblasts are commercially available, but they only replicate for a finite number of times, so they are particularly informative (and hence widely used) as primary cell culture models for the study of cellular senescence. We also sincerely appreciate all the technical questions from this reviewer, as they have helped us prepare a clearer and more complete revision. In our responses below, **new and revised figures are in bold font**.

Questions and major remarks:

- Methods:

The authors claim that BDNF belongs to the SASP, but their demonstration is still performed on human fibroblast cell lines, the authors would moderate their conclusions.

Indeed, authors would use human normal fibroblast to conclude that BDNF belong to SASP.

[AU] We apologize for being ambiguous in our original description. The entirety of the manuscript was carried out using normal primary human diploid fibroblasts (i.e., we did not employ any cell lines – transformed or immortal – at any time). We have made this point more clearly in the revised manuscript. In addition, prompted by the underlying concerns from this Reviewer, we have sought to gain more evidence in support of BDNF as a SASP component in other senescence paradigms using a variety of other primary cells. Specifically, we triggered senescence in a total of 9 senescence models (**new Fig. 2c, d**). We have now included human diploid fibroblasts exposed to ionizing radiation (IR) and hydrogen peroxide (H₂O₂), as well as other normal epithelial and endothelial cells [namely human umbilical vein endothelial cells (HUVECs), human lung small airway epithelial primary cells (HSAECs), and human kidney epithelial primary cells (HREC)], treated with etoposide. As shown in the **new Fig. 4a, b**, both *BDNF* mRNA levels (detected by RT-qPCR analysis) and BDNF secretion (detected using ELISA), increased strongly in these diverse models of cellular senescence. Again, we wish to express our gratitude to the Reviewer for his/her guidance with these comments.

Likewise, authors have to use different compound inducing senescence to rule out etoposide (PMID: 27145455, PMID: 29959561 and PMID: 25700942) or doxorubicin (PMID: 12438277) dependent mechanisms.

[AU] We completely agree with the Reviewer on this point too. By including other treatments that trigger senescence (e.g., exposure to H₂O₂ and IR), we have gained further evidence that the effects of etoposide and doxorubicin on BDNF production are not specific to these drugs. We appreciate the Reviewer's request, and his/her guidance to help us solidify these points.

Ionized-irradiation (Gy irradiation?) is not described in methods.

[AU] We regret this omission. The specifics of the treatment with Ionizing Radiation (IR) are now included in the revised Methods. Briefly, followed standard protocols, human fibroblasts were exposed to 15 Gray (Gy) of IR, returned to the incubator, and cultured for an additional 10 days.

What is the reason to use 25 or 50 etoposide in Wi38 and BJ cells respectively?

[AU] We appreciate this question. The doses of etoposide for each fibroblast type (WI-38 and BJ) were carefully titrated in a recent study from our group [Anerillas et al., 2022, *Science Advances* (PMID: 35394839)], and chosen as the optimal dose that triggered senescence without causing cell death. We have explained this fact more clearly in the revised manuscript.

Many "DNA damage" were cited by authors in the results but how was evaluated DNA damage? Notably they have not shown its specific markers such γ -H2A-X and telomere-associated DNA damage in senescent fibroblasts.

[AU] We thank the Reviewer for this valuable suggestion. As our original manuscript did not include models of replicative senescence (which cause telomere erosion), it would not have been informative to check telomere-specific damage. Instead, following the Reviewer's recommendation, we have included the suggested marker of global DNA damage, γ -H2AX. As shown (**new Fig. 6h, i**), these results indicate that the levels of γ -H2AX increase markedly in senescent cells.

SiRNA experiments have to be more detailed in particular for the time of SiRNA exposition upon etoposide. Indeed, it seems difficult to keep target extinction at 10 days.

[AU] We apologize for the scarcity of detail in our original text. We have added specifics of how we conducted the siRNA experiments (revised Materials and Methods section under 'Cell culture'). We employed RT-qPCR analysis to monitor the levels of mRNAs remaining by 10 days after transfection of siRNAs (e.g., **new Supplementary Fig. S4d and S5a**) and we consistently found that target mRNAs remained reduced for 10 days. We have used the same transfection and tracking methods in recent publications [Herman et al., 2020, *Nucleic Acids Research* (PMID: 34181735); Anerillas et al., 2022, *Science Advances* (PMID: 35394839)] and have found that silencing was similarly sustained. We hypothesize that silencing persists because senescent cells have ceased proliferation, helping siRNAs to remain concentrated, while they become progressively diluted in proliferating cells.

*Fig 1 c and Supp table 1:
Experimental conditions (concentrations and IC50 evaluation for drugs testing are not indicated).*

[AU] We appreciate the request and apologize for having missed this information in our original manuscript. We have included IC50 values (in this case, also the EC50, the concentration of a drug that gives half-maximal response) for the TrkB inhibitors in the **new Fig. 2a**.

Legend of supp Table 1 is missing.

[AU] We regret the omission and have included this legend in the revised Supplementary text.

Why Pan Trk inhibitor (GNF 5837) has superior effect to TrkB inhibitor (ANA 12) and TrkA inhibitor? Depending on its concentrations, it could interfere with PDGFR (cited by Tocris as following “Exhibits selectivity for Trk receptors over a range of kinases, with some activity at PDGFR and c-Kit (IC50 values are 0.87 and 0.91 μ M respectively)”. Concentrations are not indicated.

[AU] The Reviewer brings up a great question. We too wondered why some inhibitors were superior. We reasoned that since both compounds inhibited TrkB, the death effect caused by their presence in the medium could be related to TrkB function, which we explored in subsequent experiments. Although we cannot exclude the possibility that other targets contribute to this death phenotype, neither a PDGFR inhibitor included in the library (AC 710) (Fig. 1c) nor TrkA silencing (Fig. 3e) caused senolysis. We have specified the concentrations in the revised text.

Likewise, SRC kinase belongs to TrkB cell signaling, PP2 inhibitor presents a close profile to ANA12 by decreasing senescent cell %. These results remain not discussed in the current manuscript.

[AU] We appreciate the Reviewer’s keen observation. In a recent study [Anerillas et al., 2022, *Science Advances* (PMID: 35394839)], we report that the effectiveness of SRC inhibitors depends on their ability to inhibit integrin-dependent SRC activation in senescent cells. We cannot exclude the possibility that SRC might intersect with BDNF-TRKB survival signaling events, but our observations instead point to SRC inhibitors as primarily acting through integrin signaling. This earlier study is mentioned in the revised manuscript.

Fig 2: Method used for evaluation of cell viability is not clear. Cell counts of senescent (beta-Gal) or proliferating (BrdU) cells?

[AU] We thank the Reviewer for indicating that we needed to clarify these points. In the revised Materials and Methods section (‘Cell culture’), we now include more explicit explanation about how we measured cell viability. Briefly, images were taken at 3 random fields for each replicate from each condition, and counted cells per field using the ImageJ software. We utilized direct cell counting instead of other methods that measure cell viability based on the metabolic status of the cell, as the cell’s metabolic status is known to change with senescence. Therefore, this protocol is more robust, reproducible, and reliable, particularly when complemented with measurements of Caspase 3/7 activity to quantify differences in apoptotic cell death.

Fig 2 h and i: what are the cells? WI-38?

[AU] Yes, they are WI-38 fibroblasts. We have now indicated the type of cell studied in every legend. We thank the Reviewer for catching this omission.

-Western blots:

Fig 3: b and e: Western blots are not representative and whole blots are required with MW markers to define the full-length TrkB receptor (145 kDa) with tyrosine kinase domain and the two other truncated forms (95 kDa), TrkB-T1 and TrkB-shc without kinase domain, both detected by anti-TrkB Ab sc-136990 used in this study. Same remark for Supp Fig 3 a and c.

[AU] We appreciate this comment. We now display longer blots of TrkB in **Fig. 3a and c** (including TrkB-FL and truncated Trk, Trk-T). Given that the functional form of TrkB is full-length, we focus on TrkB-FL signals in other places in the article. The antibody cited by the Reviewer (anti-TrkB Ab sc-136990) is in fact the antibody we employed to study TrkB expression patterns. Following the Reviewer's expert advice, the Trk-T cleavage form appears to include TrkB lacking the shc domain. We thank the Reviewer for his/her guidance on this point and have included these results in the text.

As for Fig 5 a: MW western-blot of P-TrkB / TrkB are absent, the size differences are surprising. Quantification of Western-blot will be performed to assess the reproducibility of results.

[AU] We appreciate the request to clarify this point. In this experiment, we employed 'Phos-tag gels' (<https://labchem-wako.fujifilm.com/us/category/00908.html>), from FUJIFILM Wako Pure Chemical Corporation, which are formulated to accentuate the separation between a phosphorylated protein and its nonphosphorylated form. In these gels, unphosphorylated proteins and molecular weight markers migrate at the expected size; only phosphorylated proteins run much more slowly and this feature facilitates their identification. Phos-tag gels are not yet utilized widely, but their use is rapidly increasing. We have included these technical details in the revised Results and Materials and Methods sections. Given the unique properties of these gels, molecular weight markers are not very informative for the phospho forms. Densitometry quantifications have been performed and the p-TrkB/TrkB ratios have been included below the representative blots (**new Fig. 5a**).

- Immunofluorescence staining: images are not convincing.

Fig 3i: immunofluorescence staining for DPP4 is not clearly detectable, notably in contrast to the study from Kim et al. (ref. 21) from the same group that showed that DPP4 was strongly labelled in WI-38 senescent cells. There are too few cells per field, TrkB labelling is not only detected on the membrane and is also detected in the cytoplasm (while the authors indicate that the cells have not been permeabilized, therefore, a cytoplasmic diffusion in dead cells cannot be eliminated). Isotype antibody controls are not shown. Why the authors did not perform confocal microscopy analyses like Kim et al (21)? In this way, Mander's coefficient would be better to interpret colocalization with DPP4.

[AU] We appreciate these comments and realize that we did a poor job at explaining and displaying these experiments in the original manuscript. In the revision, these data are shown in the **new Fig. 3k**, and the signals from the individual fluorophores and the isotype antibody controls, as requested by the Reviewer, have been included in **new Supplementary Fig. S3f**. We cannot eliminate the possibility that cytoplasmic diffusion of the antibodies might occur in cells with compromised plasma membranes, although DPP4 is localized to the plasma membrane, so the majority of signal is expected to be at the membrane. The new immunofluorescence results support the colocalization of TrkB and the marker of senescent-cell plasma membrane DPP4.

- Fig 4:

a) The heat map shows that not only BDNF mRNA was detected but also NT3 mRNA in BJ cells (Fig 4 a) and in WI-38 RS (Supp. Fig 4 a). (Why authors have used WI-38-doxorubicin in this model, not described in Methods?) These data do not match when they are compared to those of ELISA (not detecting NT3) and array (c: detecting NT3). Furthermore, in senescent HUVECs, NT3 mRNA levels

were higher than those of BDNF. These discrepancies suggest that NT3 is also produced during senescence depending on the cell models and technical methods and that other NTs could be induced during senescence induction. This point is not discussed.

[AU] We thank the Reviewer for this request and have added details of the use of doxorubicin to the revised Methods. The Reviewer raises good points on observations that we and others have made that (1) the levels of secreted proteins do not always match the levels of mRNAs encoding these proteins, and (2) the relative levels of expressed factors (and corresponding mRNAs) change across senescence models. For example, even though *NFT3* mRNA levels appear increased in some senescence models, protein NFT3/NT3 was not always detected, likely because different protein platforms vary in sensitivity (e.g., cytokine arrays might be more sensitive than ELISA for NFT3/NT3 detection in conditioned media). We note that in the cytokine arrays (Fig. 4c), NT3 signal intensity actually decreased in senescent cells compared to proliferating cells, so it may not be informative to track *NFT3* mRNA levels in these two groups (**new Fig. 4a**). Although *BDNF* mRNA levels are consistently and strongly elevated in senescent cells, the Reviewer is correct in saying that different NTs may be induced at different points or in different senescence paradigms. Finally, it is important to be aware that, like protein, mRNA levels may be hard to detect accurately in some cases, due to low Ct values in qPCR reactions and/or low counts in RNA-seq analyses. In the revised manuscript, we briefly expand upon some of these details.

b) The ELISA procedure for BDNF and other neurotrophins measurement is not indicated. Western blotting of the supernatant of senescent cells is required to determine the type of BDNF, mature (14 kDa) or immature proBDNF (32 kDa), both detected by the sc-65514 used by the authors. This point is important given that the functions of proBDNF differ from those of its mature form (with different binding properties and coreceptor), this point requires further elucidation. Furthermore, it was not shown that the metalloproteinases associated with proBDNF cleavage (matrix metalloproteinases 3, 7...) in mature BDNF are expressed in senescent cells.

[AU] We thank the Reviewer for asking that we provide additional detail and analysis. In the revised manuscript, particularly in the revised Materials and Methods section, we offer more specifics about the ELISA analyses performed in this study. We are particularly grateful for the Reviewer's suggestions regarding BDNF processing, as they have helped us identify an interesting additional level of regulation of BDNF function. In light of the Reviewer's question, the fact that matrix metalloproteinases (MMPs) are integral components of the SASP (PMID: 20078217; PMID: 31945054), and considering that BDNF is more processed in senescent cells compared to proliferating controls (**new Fig. 4n**), we asked whether MMPs may participate in processing BDNF. By employing anti-BDNF antibody EPR1292 (Abcam, ab108319; Materials and Methods), when we inhibited MMPs with an MMP inhibitor, we observed both decreased viability and reduced BDNF processing (**Fig. 4o, p**). This interesting layer of BDNF regulation in senescence, stemming from the Reviewer's suggestion, is now included in the main manuscript (**Fig. 4n-p**).

fig 4 k: The parameters obtained from the non-relevant antibody appear to be incorrect, underestimating labelling of basal BDNF in tissues.

[AU] We appreciate the Reviewer's question, but we are not sure we totally understand what he/she means. In the original Fig. 4k (Fig. 4m in the revised manuscript), treatment of senescent cells with blocking anti-BDNF antibodies (that is, blocking BDNF in the supernatant from accessing cells) caused sensitization to cell death, as measured by increased caspase 3 and caspase 7 activities relative to IgG-treated cells. If we misunderstood the question or the Reviewer was referring to other data, kindly advise.

-Fig 6: co-staining STAT3-BDNF was not clearly superimposable (Fig 6 g). So, what is the meaning of the sentence “virtually every p-STAT3 positive cell was BDNF positive”?

[AU] We definitely see how our original statement was ambiguous. We should have been more precise by saying that while most (~60%) of the cells positive for p-STAT3 (Tyr705) were also positive for BDNF, almost all (~98%) of the cells positive for p-STAT3 were also positive for BDNF (Fig. 6l, m). We also found that in the lungs of a mouse model of senescence, almost all cells expressing p-STAT3 were also simultaneously BDNF-positive by immunofluorescence staining (**new Supplementary Fig. S6m**). These findings could be interpreted to mean that STAT3 activation is consistently accompanied by BDNF expression in senescent cells, even if the opposite is not always true (i.e., cells expressing BDNF may not always have active STAT3).

To gain alternative support for this interpretation, we performed and included new single-cell RNA-seq data (**new Fig. 6j, k**). Briefly, senescent WI-38 cells were studied by single-cell RNA-seq analysis to test if cells with different levels of *BDNF* mRNA displayed a transcriptomic profile associated with functional STAT3. After clustering the different groups (**new Supplementary Fig. S6g**), the highest levels of *BDNF* mRNA were found in clusters 4 and 5, the lowest in clusters 0 and 1 (**new Fig. 6j** and **new Supplementary Fig. S6h**). Importantly, *BDNF* mRNA levels correlated significantly with those of STAT3-regulated mRNAs, as identified by GSEA, such as *THBS1* and *FN1* mRNAs (**new Fig. 6j, k**, and **new Supplementary Fig. S6i**), and verified by ChIP-seq analysis (**new Supplementary Fig. S6h**). Moreover, by immunofluorescence staining, most of the cells expressing the highest levels of BDNF were also strongly positive for proteins THBS1 or FN1 proteins (**new Supplementary Fig. S6k**), which are translated from two STAT3-induced mRNAs in cluster 5 (**Fig. 6j**).

Fig 7: Experiments with old mice showed that BDNF is expressed by few co-expressing p16. It is difficult to understand why authors have preferentially used anti-BDNF instead of TrkB for their colocalization with p16 in mice tissues. According that TrkB/BDNF is upregulated in senescent tissues, experiments with lung, Kidney and liver with TrkB and BDNF costaining is necessary to assess their involvements during organs aging in old control mice by comparison to Trk inhibitors-treated mice.

[AU] We fully understand the Reviewer's question, we too had hoped to use an anti-TrkB antibody instead of an anti-BDNF antibody for these colocalization experiments. Unfortunately, after trying numerous antibodies on the market that are advertised as capable of detecting TrkB in mice, we did not find a single one that worked. By contrast, the anti-BDNF antibody was sensitive and reliable, so we used it as a surrogate marker. The suggested experiments to co-stain TrkB and BDNF in lung, kidney, and liver of old mice will be carried out as soon as appropriate reagents are available. Genetic approaches to use CRISPR to label TrkB-positive cells are in planning stages and will be performed as our experiments advance.

In addition, considering that senescent fibroblasts models were obtained with induced-fibroblast cell lines models, a lacking experiment would be used to validate this model with old mice fibroblasts from skin, i.e.: BDNF, TrkB and beta-Gal expressions in old control mice fibroblasts should be compared to those obtained in Trk inhibitors-treated mice.

[AU] We thank the Reviewer for this comment and again apologize that we did not clearly communicate that all our experiments had been done in primary fibroblasts (i.e., none were done in immortal or transformed fibroblasts). We regret not having been clearer on this important point. Regarding the analysis of fibroblasts from older relative to younger organisms, we had already performed the

experiment in question. A few weeks ago, we published a study of fibroblasts isolated from skin of human donors ranging in age from their early 20s to their late 80s [Tsitsipatis et al., *Aging Cell*, 2022 (PMID: 35429111)]. To our disappointment, however, the skin fibroblasts underwent senescence at comparable rates regardless of the age of the donor. We appreciate the suggestion of testing fibroblasts from mice treated with TrkB inhibitors; however, as the Reviewer likely realizes, treatment with inhibitors would cause the elimination of those senescent cells, and if they were not eliminated, the senescent cells will quickly become overtaken by proliferating cells and diluted in the primary cultures.

In Discussion: Other reports have demonstrated that normal human senescent fibroblasts secrete BDNF (Tinaburri et al., J Invest Dermatology 2021), inducing EMT in keratinocytes through TrkB activation in a paracrine loop; this is important to be discussed in the interest of inhibiting BDNF/TrkB activation in the senescent fibroblasts. The discussion could also be extended to CAF, in many cancers.

[AU] We thank the Reviewer for these suggestions and have incorporated the article by Tinaburri et al., and related points into the revised Discussion and Bibliography.

Minor issues to be addressed:

1/ The line numbers are missing which makes reviewing quite difficult.

[AU] No problem. As requested, we have added line numbers to the text version with all the revisions marked in red.

2/ The method for preparing culture supernatants for analysis of released cytokines is missing.

[AU] Thank you also for this request. We have now included a more detailed description of how the media were collected and the ELISA measurements performed.

3/ Likewise, the kit number of cytokine assay remains missing in the materials and methods section.

[AU] We have added the kit numbers to the Methods section.

4/ Methods to assess drug library also remains missing, how authors obtain percentage of cells? How this percentage is calculated?

[AU] The method for counting cells is explained above, in response to the earlier question from this Reviewer. Briefly, we captured images from 3 representative fields for each replicate from each condition, and counted cells using the software ImageJ. We chose this method because many cell counting methods rely on the metabolic status of cells, and we have empirically observed that such methods are less robust and reproducible for measuring senescent cell numbers. Instead, directly counting cells on a plate is more reliable, particularly when complemented with measurements of Caspase 3/7 activity to evaluate differences in apoptotic cell death. These considerations are indicated in the revised manuscript.

Response to the comments from Reviewer 2:

The manuscript by Anerillas et al, entitled “A BDNF-TrkB autocrine loop enhances senescent cell viability”, describes the identification of a BDNF-TrkB pathway that facilitates survival of at least certain types of senescence cells. The facts that TrkB and BDNF appear to be upregulated in senescence cells in culture and in vivo to facilitate survival of the damage cells and that BDNF colocalizes with p16 expression, making it a potential senescent cell biomarker, are both of significant interest. Thus the manuscript has the potential to make a significant contribution to the field of senescence and aging. However, there are several minor comments which the authors should try to address either experimentally or in the text of the manuscript.

[AU] We thank the Reviewer for his/her positive feedback and suggestions. In our responses below, **new and revised figures are indicated in bold font.**

Specific Questions:

1. How specific are the TrkB inhibitors GNF5837 and ANA12? Do they have any off-target effects or effects on other receptor tyrosine receptors?

[AU] We appreciate the Reviewer’s question, as inhibitors can certainly have off-target effects. GNF5837 is a potent pan-Trk inhibitor (PMID: 24900443); in assays employing Ba/F3 cells, the IC₅₀ values were 7, 9 and 11 nM for the fusion proteins Tel-TrkC, Tel-TrkB and Tel-TrkA, respectively. Approaching the micromolar range, GNF5837 can suppress PDGFR and c-Kit, with IC₅₀ values of 0.87 and 0.91 μM, respectively, far higher than what we used in this study. ANA12 is a more specific TrkB receptor antagonist (PMID: 21505263) that prevents activation by BDNF via two sites, a high-affinity site (IC₅₀ = 45.6 nM) and a low-affinity site (IC₅₀ = 41.1 μM), and does not appear to affect TrkA or TrkC function in neurite outgrowth assays. Besides including these citations, we have strengthened the characterization of these drugs throughout the manuscript to support their specificity on TrkB.

2. Dasatinib, a “dirty” tyrosine kinase inhibitor, is a known senolytic. Is there any evidence that dasatinib inhibits TrkB?

[AU] We thank the Reviewer for this question, as Dasatinib is indeed a wide-spectrum inhibitor of tyrosine kinases, and a senolytic drug (typically used in combination with Quercetin). Unfortunately, in our study, we did not find any evidence that Dasatinib directly inhibited TrkB. Dasatinib does inhibit the kinase SRC, as we recently found [Anerillas et al., 2022, *Science Advances* (PMID: 35394839)], and hence it could indirectly suppress signaling initiated through TrkB, should a downstream component of TrkB signaling require SRC activity. We mention these possibilities in the revised manuscript.

3. Do the TrkB inhibitors work as senolytics on other senescent cell types such as HUVECs, HAECs, astrocytes, preadipocytes or immune cells?

[AU] We appreciate and fully agree with this important question. We have now included several other models of senescence, such as human umbilical vein endothelial cells (HUVECs), human lung small airway epithelial primary cells (HSAECs), and human kidney epithelial primary cells (HREC)s. We also expanded our survey to other models of senescence triggered by hydrogen peroxide (H₂O₂), ionizing radiation (IR), and replicative exhaustion (all confirmed by increased SA-β-Gal activity and decreased BrdU incorporation; **new Supplementary figs. 2b-d**). Even though the optimal doses causing senolysis varied across the different models of senescence tested, TrkB inhibitors both reduced

viability and increased Caspase 3/7 activity in all of them (**new Fig. 2c, d**); GNF 5837 had the most consistent effect across the cell types and triggers studied. We thank the reviewer for this valuable request.

4. The EC50s for the TrkB inhibitors should be provided. Also, it should be determined if the IC50 for TrkB kinase inhibition is the same as the EC50 for senolytic activity.

[AU] We also appreciate these questions. We have now included EC50 values for the inhibitors (**Fig. 2a**). Unfortunately, we were not able to compare if the IC50 for TrkB inhibition is the same as the EC50 for senolytic activity, as it was not reported if these inhibitors impede TrkB phosphorylation at a certain residue or they simply prevent downstream signal transduction. If the Reviewer feels strongly that we should look further at this parameter, we would be very interested in his/her instructions. Perhaps it helps to add that using ERK5 phosphorylation as a readout of BDNF-TrkB activation in senescent WI-38 cells, we confirmed that all the TrkB inhibitors tested decreased ERK5 phosphorylation at the doses causing senolysis (**Supplementary Fig. S5g**).

5. It is unclear what the authors define as early and late senescence. Also, Line-1 is a better marker of late senescence than p16, which is elevated in early senescence.

[AU] We appreciate the request to say more about early and late senescence, and have included two relevant articles in the revised text (PMID: 28822679, PMID: 30602768). We have specified the days after etoposide treatment at which the samples were collected in the figure legends (2 days for early senescence, 8 days for late senescence). We have also assessed the expression levels of the *LINE1* RNA throughout senescence in several of the fibroblast cultures and we observed that the patterns did not follow the same dynamics as that reported previously (PMID: 30728521) for replicative senescence (**Fig. R1 for reviewers**).

Fig. R1 for Reviewers. Human diploid fibroblasts WI-38, BJ, and IMR-90 were treated with etoposide (to achieve etoposide treatment-induced senescence, ETIS) for the times indicated, and the abundance of *LINE1* RNA was assessed by RT-qPCR analysis and normalized to the levels of *GAPDH* mRNA.

6. Why were 1 month time points chosen for treatment of old mice with the TrkB inhibitors? Other groups have dosed more frequently with senolytics since the burden of SnCs increases several weeks post senolytic treatment. How long after the last treatment were the mice sacrificed?

[AU] We appreciate this important question. We designed the TrkB inhibitor experiment carefully after considering published literature, especially a recent paper on the role of the *Prfl* gene on the accumulation of senescent cells in mice (PMID: 30575733). We would have liked to have included other frequencies of treatment with inhibitors, as well as different times of collection after treatment, but we had to make decisions to stay within constraints of time and cost for this project.

7. The results suggest that BDNF, if it indeed is a SASP factor, should increase in the plasma with aging. Also expression of BDNF should go up in tissues and cell types in vivo. The inclusion of at least some evidence for increased BDNF expression at the RNA level in tissues and at the protein level in serum should be included or at least mentioned.

[AU] We thank the Reviewer for this suggestion. We measured BDNF in different tissues, but we were not able to detect significant changes among the groups tested in **Fig. 7**, except for one mouse that displayed high BDNF levels in serum, probably due to an unrelated cause. In sum, the effect may be local, and BDNF may be taken up quickly, and it may not be detectable on a global level in circulation.

8. Antibodies against murine p16INK4a are notoriously bad. The authors need to provide Supplementary data that the antibody used for the in vivo analysis of senescent cells indeed recognizes p16 such as through the use of p16 deficient cells or mice.

[AU] As the Reviewer notes, many antibodies that recognize mouse p16 are of poor quality, although in our experiments we employed an antibody that was recently validated (PMID: 30575733). Nonetheless, we followed the Reviewer's advice to confirm that the signal truly corresponded to p16 in tissues from old mice. We pursued two approaches. In one, Dr. Darren Baker kindly sent us tissues from WT and p16-null mice for us to test; unfortunately, none of the tissues showed p16-positive staining because the tissues were from mice that were relatively young (12 months old). In the other, we employed a knock-in mouse strain recently published (PMID: 30683717), that contains a tdTomato reporter in place of one of the alleles of the *p16Ink4a* gene. As these mice were older (>24 months old), we were able to observe co-staining of cells by double immunofluorescence with anti-tdTomato antibody (the tomato fluorescence itself was not strong enough for detection) and the anti-p16 antibody used in the manuscript. Importantly, most cells positive for either of the two antibodies used were positive for both antibodies (**new Supplementary Fig. S7a**), supporting the validity of the anti-p16 antibody as a reliable way to detect p16 expression in mice. We appreciate the Reviewer's encouragement to include this information in the revised manuscript.

9. Is TrkB, which appears to be upregulated at the protein level and present on the cell surface, a better marker than BDNF for senescent cells? What percent of p16+ cells in vivo are TrkB+.

[AU] Again, we thank the Reviewer for this question. As explained above in our response to Reviewer 1, anti-TrkB antibodies were not adequate for immunofluorescence. This is likely a widespread problem in the field, given that there are not many anti-TrkB antibodies commercially available for histology and that most groups studying TrkB *in vivo* make transgenic reporter mouse models to avoid having to use antibodies. In addition, since TrkB is a membrane protein and p16 is a nuclear protein, harsh permeabilization procedures needed to visualize p16 may cause the elimination of signals coming from plasma membrane proteins like TrkB. By contrast, the antibody we used to detect BDNF is robust and is commonly used for histological detection. We have calculated the number of p16-positive cells that are also BDNF-positive and it appears that ~50% of p16-positive cells in old mice in kidney, lung, and liver are also BDNF-positive (**new Fig. 7a**). In the revised manuscript, we discuss the current technical limitations and describe these new important results.

10. There are numerous panels of senescence and SASP factors that have been reported including CellAge and Sen_Mayo. However, BDNF is not a component of these gene sets, possibly because the increase in expression of BDNF in senescent cells is only 2 fold at best. The authors need to discuss why certain senescent/SASP gene sets, used to identify and characterize senescent cells, haven't picked up

BDNF. Also, how does a two fold increase in BDNF mRNA lead to such high expression of protein so that it allows for identification of senescent cells by IF?

[AU] We appreciate the Reviewer's raising these interesting points. We too were surprised by the unexpected discovery that BDNF, a neuronal factor, was secreted by senescent fibroblasts. In this revision, we have included several more senescence models, including those achieved by HUVECs (human umbilical vein endothelial cells), HSAECs (human lung small airway epithelial cells), and HRECs (human renal epithelial primary cells) rendered senescent by etoposide treatment, and have also included other forms of senescence implementation besides treatment with etoposide; now we also have treatment with ionizing radiation (IR-induced senescence or IRIS), treatment with H₂O₂ (oxidative stress-induced senescence or OSIS), and replicative senescence (RS) reached by exhaustion of cell division. We found that while the increases were not dramatic, senescent cells consistently and significantly augmented the production of *BDNF* mRNA and the secretion of BDNF (**new Fig. 4a, b**). Regarding the lack of previous reports on BDNF as a signature protein associated with the senescence transcriptome or proteome, most of these studies focus on mRNAs and proteins that are *overall more abundant* and *more differentially expressed* in senescence. Given this limitation, molecules that were less abundant or changing less with senescence were likely missed from these high-throughput global panels.

11. The regulation of BDNF at different levels of DNA damage is of interest but does raise questions regarding how it is regulated by p53, STAT3 and additional pathways. Is a similar differential regulation observed with replicated senescence, senescence induced by inflammatory factors, oncogene-induced senescence and/or IR. Although a full examination of this question would be difficult, analysis of BDNF expression following a different inducer of senescence would be appropriate.

[AU] We fully agree with the Reviewer and appreciate his/her suggestion. We studied senescence induced by oxidative stress (H₂O₂; OSIS), distinct from etoposide but also easily testable at a range of doses. Notably, we observed the same patterns, with only sublethal (senescence-causing) doses of H₂O₂ leading to increased BDNF production, while higher (lethal) doses of H₂O₂ triggered cell death and did not increase BDNF production (**new Fig. 6i**). In addition, we performed single-cell RNA-seq analysis of WI-38 fibroblasts rendered senescent by treatment with etoposide, and observed that the clusters of cells expressing highest levels of *BDNF* mRNA were also expressing the highest levels of STAT3-regulated transcripts such as *FN1* or *THBS1* mRNAs (**new Fig. 6j,k**).

Response to the comments from Reviewer 3:

In the current study, Anerillas et al. reported a novel Senescent Cell Anti-Apoptotic Pathways (SCAPs) that senescent cells survive through a BDNF-TrkB autocrine loop. They first screened a group of compounds affecting MAPK pathways and then discovered that inhibitors targeting the TrkB receptor had the maximum senolytic effects. They further demonstrated that senescent cells highly produce BDNF binding to TrkB to activate the ERK pathway, which upregulates known SCAPs, Bcl-W. The discovery of novel SCAPs is exciting and important, extending the therapeutic potentials of senolytics. However, I have several main concerns:

[AU] We thank the Reviewer for his/her support and feedback. In our responses below, **new/revised figures are in bold font**.

1. The authors used two types of fibroblast cells to test the BDNF-TrkB autocrine loop in this study. To test whether the BDNF-TrkB autocrine loop is cell type-dependent, can the authors assess this SCAP in other types of cells such as endothelial cells or mouse MEFs?

[AU] We appreciate the Reviewer's request, which was also identified as a deficiency by other Reviewers. In this revision, we have incorporated additional cell types and senescence triggers, now totaling 9 models of senescence (**new Fig. 2c, d, Fig. 4a, b**). The new cell models include human umbilical vein endothelial cells (HUVECs), lung small airway epithelial primary cells (HSAECs), and kidney epithelial primary cells (HREC), all rendered senescent by exposure to etoposide; the new survey is complemented by senescence triggered in fibroblasts by etoposide, oxidative stress elicited by treatment with H₂O₂, ionizing radiation (IR), and replicative exhaustion (RS). Across these models (ETIS, OSIS, IRIS, and RS, respectively), we have seen consistently that the levels of *BDNF* mRNA and secreted BDNF increase with senescence (**new Fig. 4a, b**), and have observed the effectiveness of TrkB inhibitors in reducing the viability of senescent cells specifically (**new Fig. 2c, d**). In addition, silencing BDNF production in HUVEC cells before triggering senescence rendered similar results as those observed after silencing BDNF production in WI-38 cells, namely that cells encountered enhanced cell death in response to etoposide treatment (**new Fig. 4i**). We thank the Reviewer for requesting that we expand our experiments to include these important additional models, and we have modified the text to incorporate these results.

2. In the previous publication (PMID: 31945054), the senescence-associated secretory phenotype (SASP) of senescent fibroblast cells was fully characterized using a comprehensive proteomic approach. However, BDNF was not found highly secreted by senescent cells. The discrepancy between studies might be caused by the induction of senescence, X-irradiation, and inducible RAS overexpression in the previous study, and chemo-drug (etoposide) was used to induce senescence in the current study. It raises whether the BDNF-TrkB autocrine loop solely depends on the etoposide induction. Can the authors use X-irradiation or Oncogene induce senescence to test the BDNF-TrkB autocrine loop?

[AU] We thank the Reviewer for bringing up another key question. In response to this request, we have triggered senescence with three additional agents, ionizing radiation (IR) as suggested by the Reviewer, as well as by exposure to H₂O₂, and by replicative exhaustion. As shown (**new Fig. 4a, b**), both *BDNF* mRNA and secreted BDNF are elevated across all 9 senescence models. As explained above while responding to a similar query from Reviewer 2, the fact that previous studies did not identify BDNF as a member of the SASP might be due to low coverage of proteomic approaches, low abundance of BDNF compared to other SASP factors, and/or a focus on proteins that were more differentially abundant. To the Reviewer's specific question about the BDNF-TrkB autocrine loop, we tested it in senescence paradigms triggered by H₂O₂ or IR; as shown in the **new Fig. 4i**, the newly included senescence

programs also rely on BDNF production to sustain viability. In the revised text, we discuss the expansion of the BDNF-TrkB paradigm to other models of senescence and highlight the pro-survival influence of signaling through BDNF-TrkB across senescence paradigms.

3. In Fig. 1 and 2, the TrkB inhibitors showed the most effective senolytic activities among other compounds. I noticed that EC50 of TrkB inhibitors is 10 times lower than known senolytics, ABT-263/ABT-737. Does this suggest additional independent signaling contributing to senescent cell survival? If yes, the authors should discuss it in the discussion section.

[AU] We appreciate the Reviewer's keen observation, as well as his/her question and advice. In the revised Discussion, we have included text explaining that the current senolytics can have side effects due to their ability to inhibit pro-survival, BCL2-related proteins and thus they reduce viability globally across cell types. Given that TrkB inhibitors diminish survival of senescent cells at lower doses and appear to operate mainly through BCL2-independent pathways, the two interventions could be exploited synergistically to potentiate the death of senescent cells. As our work progresses, we hope to investigate this possible synergy in dedicated studies.

4. The authors used the term "early senescence" and "late senescence" to distinguish the changes of Bcl-2 families during senescence in the cultured cells. Can the authors clarify the "early" and "late" stages? Does early senescence exist in vivo? Or is this an observation of the pre-senescence stage? Maybe a time course study is more helpful.

[AU] This important question was also raised by Reviewer 2. In response to both Reviewers, we include in the revised text earlier studies that address the dynamic nature of senescence (PMID: 28822679, PMID: 30602768). We now discuss "early" and "late" senescence more explicitly, and we underscore the growing interest by scientists in understanding the molecular changes that mark this progression. While the genes and signaling kinetics leading to the implementation of senescence are not yet fully known, we and others are deeply interested in identifying them systematically. As these markers become better defined, we will be able to interrogate early senescence in tissues *in vivo*. In fact, our team is gearing up to systematically investigate senescent cell dynamics in culture and *in vivo*. We anticipate that understanding senescence dynamics in tissues will be particularly informative to understand the impact of senescent cells in specific tissue environments; for example, whether senescent cells might elicit a pro-fibrotic or a pro-inflammatory influence.

5. In the experiment of knocking down BDNF to test senescent cell viability, it would be more convincing if the authors could show adding soluble BDNF to prevent senescent cell death.

[AU] We thank the Reviewer for the brilliant suggestion of this experiment, which could provide evidence for the notion that BDNF helps support cell viability in senescence. As shown in the **new Fig. 4j**, ectopic addition of BDNF at the approximate concentration measured in senescent WI-38 fibroblasts (determined by ELISA to be ~200 pg/ml) significantly improved the survival of WI-38 fibroblasts in which BDNF had been silenced. These results have been included and discussed.

6. It is known that the survival of senescent fibroblast relies on Bcl-xl, Bcl-2, and Bcl-w. In Fig. 5g, it would be clearer if the authors could show the protein expression of Bcl-2 and Bcl-xl after knocking down NTRK2 and BDNF.

[AU] We are grateful for this valuable suggestion. We have done the experiments requested and have included the data in the **new Fig. 5g**; as shown, there are no big changes in the levels of BCLW, BCLxL

and BCL2. In light of these data, we discuss in the revised text that BDNF-TrkB signaling relies at least in part on BCL2L2 function as a pro-survival effector in senescent cells.

7. The authors speculate that the p-STAT3 pathway primes the production of BDNF. If so, will Jak/STAT inhibitor induce senescent cell death after etoposide induction?

[AU] The Reviewer makes an interesting suggestion. Contrary to our prediction (and the Reviewer's prediction), inhibiting STAT3 did not reduce the viability of cells in which senescence was triggered by etoposide treatment (Supplementary Fig. S6f). This observation might indicate that even though STAT3 promotes BDNF production, other pathways may also induce BDNF production during the implementation of senescence, and sensitization to apoptosis can only be achieved by direct BDNF inhibition. Recently, we identified an early dichotomy between senescence and apoptosis and discovered that an epithelial-to-mesenchymal transition (EMT) program favored senescence [Anerillas et al., 2022, *Science Advances*, PMID: 35394839]. In this regard, BDNF has also been linked to EMT in several situations (as we mention briefly in the revised Discussion) and in this report (**new Supplementary Fig. S6j**). Therefore, and as the EMT process is known to result from the convergence of several pathways, inhibition of STAT3 could be compensated by other EMT-promoting programs.

8. In in vivo study, the authors showed the reduction of senescence markers in multiple tissues. It would be more convincing if the authors could provide more evidence of improvement of age-related phenotypes in the treated animals. Also, the cognitive test is essential to show no detrimental effects on CNS.

[AU] We appreciate the Reviewer's comments and have addressed them in several ways. First, we performed Picro-Sirius red staining to visualize connective tissue (mainly collagen I and III fibers) and assess if there are differences in fibrosis. As shown in the **new Supplementary Fig. S7f**, in the different tissues studied in this manuscript, mice treated with TrkB inhibitors displayed less fibrosis-related staining. Second, combining this assay of fibrosis with serum markers of kidney function, we observed that mice treated with TrkB inhibitors showed reduced fibrotic areas in the kidney, along with improved renal function, as determined by measuring urea and creatinine levels in serum (**new Fig. 7i,j**). Finally, although neither GNF 5837 nor PF 06273340 cross the blood-brain barrier, and hence we expected no detrimental cognitive side effects, we did analyze brain tissue for apoptosis using TUNEL (terminal deoxynucleotidyl transferase dUTP nick-end labeling) and assessed neuronal markers. As shown in the **new Supplementary Fig. S7g, h**, these assays revealed that treatment of mice with TrkB inhibitors did not cause reduced neuronal density nor noticeable neuronal death in the brain cortex.

9. In Fig. 7d, one SASP factor, GDF15, was shown decreased by TrkB inhibitors. What about other well-known SASP factors? This is very important because it may indicate that TrkB inhibitors only target a specific type of senescent cells.

[AU] We appreciate the Reviewer's comment. With advice from our colleague (Nathan Basisty, PMID: 31945054), we assessed other SASP factors changing in the human secretome with age. However, as shown in Fig. 7c and Supplementary Fig. 7c, in mouse serum samples the only factor showing significant changes in the conditions studied was GDF15, while other factors identified in the human study (PMID: 31945054), such as PAI-1 or TIMP-1, showed the expected trends (as did *Il6* and *Il1b* mRNAs in Fig. 7c) but did not reach significance across any groups, including when comparing young control mice and old untreated mice. Work is underway to perform more comprehensive proteomic analysis similar to what was reported in PMID: 31945054 in collaboration with N. Basisty, but these studies will require 1-2 years' time.

Minor suggestions:

1. *In the Fig. legend of Fig. 1c, the concentration of compounds is missing.*

[AU] We apologize for the omission and have included the concentrations.

2. *I noticed that the authors used DPP4 as a marker of senescent cells in Fig. 3. p16 is the most used senescence marker. It would be more convincing if the authors could provide the protein expression of p16 in etoposide-induced senescent cells and show how the protein level of TrkB and BDNF changes along with p16.*

[AU] We appreciate this request and realized that we were not clear in our original description. We used DPP4 as a *membrane* marker of senescent cells, as identified previously (PMID: 28877934) and used by others afterwards (e.g., PMID: 33446552). The goal of the experiment in Fig. 3 was to identify TrkB enriched on the membrane of senescent cells, but we failed to make this point explicitly in the text. Given that p16 is a nuclear protein, it could not be used for this purpose, since visualizing p16 would have required harsher permeabilization methods that would have compromised the detection of proteins present in the plasma membrane. Instead, DPP4 was a robust senescent marker in these localization studies focused on the plasma membrane.

3. *In Fig. 4d, the graph to show the quantification of intensities of BDNF is missing.*

[AU] Thank you for catching this. In the revised manuscript we have added the quantification of BDNF intensities (**new Fig. 4e, f**).

4. *In Fig. 7a, the antibody validation for the immunofluorescence staining of p16 in mouse long and liver is required.*

[AU] We appreciate the Reviewer's request to validate the p16 staining, echoing a request from Reviewer 2. Although the antibodies to recognize p16 in mice are not great, we used an antibody that was recently validated (PMID: 30575733). To confirm that the signal we observed corresponded to p16, we used a knock-in mouse strain recently published (PMID: 30683717) that contains a tdTomato reporter in the place of one of the alleles of the *p16Ink4a* gene. Importantly, in mice older than 24 months old, we observed co-staining of cells by double immunofluorescence with anti-tdTomato antibody (this strategy was employed because the Tomato fluorescence on its own was not strong enough for detection) and the anti-p16 antibody used in the manuscript. We noted that most cells positive for either of the two antibodies used were positive for both antibodies (**new Supplementary Fig. S7a, b**), supporting the validity of the anti-p16 antibody as a reliable way to detect p16 expression in mice. We appreciate the Reviewer's encouragement to include this information in the revised manuscript.

5. *A detailed description of statistical analysis is missing in both the Fig. legend and Method section.*

[AU] We have included all the details on our statistical analyses for this manuscript in the Figure Legends and in the Materials and Methods section. Briefly, we used Standard Deviation (S.D.), a minimum of three data points per measurement, and Student's *t*-test to determine significance.

REVIEWERS' COMMENTS

Reviewer #1 (Remarks to the Author):

The modifications made by the authors are in response to my questions and comments. I agree with this modified version of the manuscript.

Reviewer #3 (Remarks to the Author):

The revised manuscript has been dramatically improved. I'm satisfied with all responses. The authors provided extensive additional evidence to support the central hypothesis. I believe that the finding of the novel survival pathway of senescent cells will significantly impact the field. I have two remaining questions/comments according to the second reviewer:

Question # 7: BDNF can be uptake quickly, but at least the mRNA level of BDNF should be detectable. At least the change of BDNF at the mRNA level can be shown here to answer this question.

Question # 10: The authors' response partially answered this question. I believe that the 2nd reviewer meant the differential expression of BDNF between SnCs and healthy cells is moderate, which is unlikely to be distinguished by IF as a biomarker. It also raises concerns about the side effects of using BDNK as a drug target. Hence, the authors need to discuss this concern in the discussion.

Reviewer #1 (Remarks to the Author):

The modifications made by the authors are in response to my questions and comments. I agree with this modified version of the manuscript.

[AU] We appreciate the positive response from Reviewer 1.

Reviewer #3 (Remarks to the Author):

The revised manuscript has been dramatically improved. I'm satisfied with all responses. The authors provided extensive additional evidence to support the central hypothesis. I believe that the finding of the novel survival pathway of senescent cells will significantly impact the field.

[AU] We thank Reviewer 3 for his/her positive comments, as well as for helping to evaluate our responses to Reviewer 2.

I have two remaining questions/comments according to the second reviewer:

Question # 7: BDNF can be uptake quickly, but at least the mRNA level of BDNF should be detectable. At least the change of BDNF at the mRNA level can be shown here to answer this question.

[AU] We appreciate the chance to further explain this point and apologize for not having included these data in the original revision. We had previously checked *Bdnf* mRNA (as well as *Gdf15* mRNA, a positive control for senescent cells) in whole-tissue extracts from the Young and Old mice included in the study. The results, shown below (**Data for Reviewers, Figure 1**), indicate that *Bdnf* mRNA was not significantly elevated when measuring it in whole tissues (kidney, lung, and liver), possibly because the percentages of senescent cells in typical organs are low, and/or senescent cells may not express *Bdnf* mRNA at all stages of senescence (e.g., *Bdnf* mRNA levels might be low early in senescence, as observed in **Figure 4g**). The fact that *Gdf15* mRNA was also generally unchanged suggests that senescent cells are not easy to study at the whole-tissue level. Experiments are underway in our laboratory to identify senescent cells in different organs using single-cell analysis, and we will examine *Bdnf* mRNA using this approach. We hope to report these results in the near future, as our studies continue.

Data for Reviewers, Figure 1. Total RNA was prepared from kidney, lung, and liver from young (3 months old) and old (24 months old) mice, and the levels of *Bdnf* mRNA and *Gdf15* mRNA were measured by RT-qPCR analysis, and normalized to the levels of *Actb* mRNA measured in the same organs. The graphs reflect individual data points measured from different mice; values are displayed as the means \pm SD; significance (* $p < 0.05$) was determined by using two-tailed Student's t-test. 'ns', not significant.

Question # 10: The authors' response partially answered this question. I believe that the 2nd reviewer meant the differential expression of BDNF between SnCs and healthy cells is moderate, which is unlikely to be distinguished by IF as a biomarker. It also raises concerns about the side effects of using BDNF as a drug target. Hence, the authors need to discuss this concern in the discussion.

[AU] We appreciate the recommendation that we address these points more clearly and have incorporated them into the revised Discussion. Using the detection techniques available to us for this study (i.e., ELISA, immunofluorescence, RT-qPCR analysis), we agree that BDNF does not appear to be an ideal biomarker or therapeutic target for senescent cells. As we advance in our single-cell experiments and spatial detection methods, we hope to gain more precise information regarding the usefulness of BDNF as a marker and therapeutic target in aging and in situations where senescent cells may be detrimental.